# RNA-binding proteins distinguish between similar sequence motifs to promote targeted deadenylation by Ccr4-Not

Michael W Webster, James AW Stowell, Lori A Passmore*

MRC Laboratory of Molecular Biology, Cambridge, United Kingdom

**Abstract** The Ccr4-Not complex removes mRNA poly(A) tails to regulate eukaryotic mRNA stability and translation. RNA-binding proteins contribute to specificity by interacting with both Ccr4-Not and target mRNAs, but this is not fully understood. Here, we reconstitute accelerated and selective deadenylation of RNAs containing AU-rich elements (AREs) and Pumilio-response elements (PREs). We find that the fission yeast homologues of Tristetraprolin/TTP and Pumilio/Puf (Zfs1 and Puf3) interact with Ccr4-Not via multiple regions within low-complexity sequences, suggestive of a multipartite interface that extends beyond previously defined interactions. Using a two-color assay to simultaneously monitor poly(A) tail removal from different RNAs, we demonstrate that Puf3 can distinguish between RNAs of very similar sequence. Analysis of binding kinetics reveals that this is primarily due to differences in dissociation rate constants. Consequently, motif quality is a major determinant of mRNA stability for Puf3 targets in vivo and can be used for the prediction of mRNA targets.
DOI: https://doi.org/10.7554/eLife.40670.001

## Introduction

Shortening or removal of mRNA poly(A) tails (deadenylation) represses gene expression in eukaryotes. The Ccr4-Not complex is a conserved deadenylase that can initiate cytoplasmic mRNA decay and reduce translation by releasing poly(A)-binding protein (Pab1/PABPC1) (*Parker, 2012*; *Tucker et al., 2001*). Ccr4-Not contains seven core subunits, including two poly(A)-specific exonucleases, Ccr4/CNOT6/CNOT6L and Caf1/Pop2/CNOT7/CNOT8 (*Wahle and Winkler, 2013*). It can be specifically recruited to mRNAs by RNA-binding proteins and the RNA-induced silencing complex (RISC). Ccr4-Not recruitment can also be modulated by codon optimality and covalent modifications of the RNA sequence (*Du et al., 2016*; *Eulalio et al., 2009*; *Wahle and Winkler, 2013*; *Webster et al., 2018*). Diverse biological processes depend on deadenylation, including embryogenesis, germline maintenance, cell proliferation, and attenuation of the immune response (*Carballo et al., 1998*; *Leppek et al., 2013*; *Subtelny et al., 2014*; *Yamaji et al., 2017*).

RNA-binding proteins such as Pumilio/Puf and Tristetraprolin/TTP can recognize sequence motifs in mRNAs, and also bind Ccr4-Not to target those RNAs for deadenylation (*Wahle and Winkler, 2013*). The set of transcripts targeted for repression and decay by a given RNA-binding protein is therefore defined by the distribution of its binding motifs across the transcriptome, and often includes groups of mRNAs that encode functionally related proteins. This generates RNA regulatory networks, structured such that a single protein can regulate the expression of numerous genes (*Joshi et al., 2011*; *Keene, 2007*; *Lapointe et al., 2017*; *Wilinski et al., 2015*). Targets of RNA-binding proteins often include factors that regulate gene expression themselves, thereby forming gene expression cascades (*Barckmann and Simonelig, 2013*; *Lapointe et al., 2017*; *Shyu et al.,*

*For correspondence:
passmore@mrc-lmb.cam.ac.uk

Competing interests: The authors declare that no competing interests exist.

**eLife digest** When a cell needs to make a particular protein, it first copies the instructions from the matching gene into a molecule known as a messenger RNA (or an mRNA for short). The more mRNA copies it makes, the more protein it can produce. A simple way to control protein production is to raise or lower the number of these mRNA messages, and living cells have lots of ways to make this happen. One method involves codes built into the mRNAs themselves. The mRNAs can carry short sequences of genetic letters that can trigger their own destruction. Known as "destabilising motifs", these sequences attract the attention of a group of proteins called Ccr4-Not. Together these proteins shorten the end of the mRNAs, preparing the molecules for degradation. But how does Ccr4-Not choose which mRNAs to target?

Different mRNAs carry different destabilising motifs. This means that when groups of mRNAs all carry the same motif, the cell can destroy them all together. This allows the cell to switch networks of related genes off together without affecting the mRNAs it still needs. What is puzzling is that the destabilising motifs that control different groups of mRNAs can be very similar, and scientists do not yet know how Ccr4-Not can tell the difference, or what triggers it to start breaking down groups of mRNAs. To find out, Webster et al. recreated the system in the laboratory using purified molecules.

The test-tube system confirmed previous suggestions that a protein called Puf3 forms a bridge between Ccr4-Not and mRNAs. It acts as a tether, recognising a destabilising motif and linking it to Ccr4-Not. Labelling different mRNAs with two colours of fluorescent dye showed how Puf3 helps the cell to choose which to destroy. Puf3 allows Ccr4-Not to select specific mRNAs from a mixture of molecules. Puf3 could distinguish between mRNAs that differed in a single letter of genetic code. When it matched with the wrong mRNA, it disconnected much faster than when it matched with the right one, preventing Ccr4-Not from linking up.

The ability to destroy specific mRNA messages is critical for cell survival. It happens when cells divide, during immune responses such as inflammation, and in early development. Understanding the targets of tethers like Puf3 could help scientists to predict which genes will switch off and when. This could reveal genes that work together, helping to unravel their roles inside cells.

DOI: https://doi.org/10.7554/eLife.40670.002

*1991*; *Wells et al., 2012*). Identification of the mRNAs within these networks allows a better understanding of gene expression and the roles of RNA-binding proteins that regulate them.

Recently, three sequence elements that strongly promote mRNA destabilization were identified in an unbiased screen in zebrafish embryogenesis: the Pumilio-response element (PRE), the AU-rich element (ARE) and the miR-430 seed sequence (*Rabani et al., 2017*). PRE- and ARE-mediated mRNA destabilization is conserved from yeast to human but the set of mRNAs that contain these motifs are different across species (*Brooks and Blackshear, 2013*; *Wells et al., 2015*; *Wickens et al., 2002*; *Wilinski et al., 2017*). Consequently, the biological processes regulated by orthologous RNA-binding proteins are diverse across organisms.

The PRE sequence motif is recognized by a conserved RNA-binding domain found within the Pumilio protein family. A direct interaction between the Pumilio RNA-binding domain and the CNOT7/Caf1 exonuclease has been reported (*Van Etten et al., 2012*), but it is unknown if this is the only site of interaction on Ccr4-Not. The PRE that interacts with human PUM1 and PUM2 (UGUA-NAUA) is found in mRNAs encoding proteins involved in signaling pathways and neuronal processes (*Bohn et al., 2018*). In *Saccharomyces cerevisiae* (*Sc*), there are six Pumilio proteins. Of these, *Sc*Puf3 is most closely related to homologs in higher eukaryotes (*Wickens et al., 2002*). The RNA motif recognized by *Sc*Puf3 resembles that of the human PUM proteins but, additionally, cytosine is preferred in the −1 and −2 positions (*Gerber et al., 2004*; *Kershaw et al., 2015*; *Lapointe et al., 2015*; *Wickens et al., 2002*; *Zhu et al., 2009*). The closest homolog to human PUM in *Schizosaccharomyces pombe* (Puf3) also contains residues that define this additional selectivity pocket, and is therefore predicted to bind sequences containing an upstream cytosine (*Qiu et al., 2012*).

*S. cerevisiae* Puf3 is a key regulator of mitochondrial function (*Lee and Tu, 2015*; *Saint-Georges et al., 2008*). Consistent with this, its mRNA targets encode proteins localized to the mitochondria and involved in the oxidative phosphorylation pathway (*Gerber et al., 2004*;

*Kershaw et al., 2015*; *Lapointe et al., 2015*; *Lapointe et al., 2018*). Many transcripts proposed to be regulated by *Sc*Puf3 do not, however, show a clear link to this role (*Kershaw et al., 2015*). It is therefore unclear whether the function of *Sc*Puf3 is limited to mitochondrial regulation. Puf3 homologs in other organisms (including *S. pombe*) do not necessarily regulate mitochondrial function (*Hogan et al., 2015*).

The nine-nucleotide ARE motif (UUAUUUAUU) was identified in unstable mRNAs encoding cytokines and lymphokines in human cells (*Caput et al., 1986*; *Chen and Shyu, 1995*). This sequence is recognized by tandem zinc finger (TZF) proteins including tristetraprolin (TTP). TTP attenuates immunological responses by binding directly to Ccr4-Not via the CNOT1 and CNOT9 subunits, targeting ARE-containing transcripts for degradation (*Brooks and Blackshear, 2013*; *Bulbrook et al., 2018*; *Fabian et al., 2013*). Phosphorylation of TTP occurs as part of the p38 MAPK pathway, and this disrupts Ccr4-Not recruitment (*Marchese et al., 2010*). The *S. pombe* Zfs1 protein is homologous to TTP, and recognizes the same RNA motif (*Cuthbertson et al., 2008*; *Wells et al., 2015*). An interaction between Zfs1 and Ccr4-Not has not been characterized in fission yeast and the Ccr4-Not-interacting amino acid sequences of TTP are not clearly conserved in Zfs1.

Understanding the molecular basis of accelerated deadenylation has been limited by the lack of a biochemical system containing purified components that reconstitutes this process. Previous studies have shown that a purified domain of the budding yeast Pumilio protein Mpt5 stimulates the activity of immunoprecipitated Ccr4-Not (*Goldstrohm et al., 2006*), and that isolated Caf1 is stimulated by addition of purified BTG2 and PABPC1 (*Stupfler et al., 2016*). We recently purified the complete seven-subunit *S. pombe* Ccr4-Not complex after overexpression of the subunits in insect cells (*Stowell et al., 2016*). Biochemical assays revealed that this recombinant complex was substantially more active than the isolated nuclease enzymes (*Stowell et al., 2016*; *Webster et al., 2018*). Co-expression of the conserved subunits of Ccr4-Not with Mmi1, an RNA-binding protein found in fission yeast, generated a complex that deadenylated Mmi1-target RNAs more rapidly than non-target RNAs (*Stowell et al., 2016*).

Here, we reconstitute accelerated and selective deadenylation of PRE- and ARE-containing RNAs using recombinant *S. pombe* proteins. We find that Puf3 and Zfs1 act as molecular tethers capable of inducing accelerated and RNA-selective deadenylation by Ccr4-Not in vitro. Biochemical and biophysical analyses of Puf3 binding to RNA reveal a high degree of sequence selectivity. Correspondingly, in *S. cerevisiae* RNA motif quality is a critical determinant of the RNAs stably bound by Puf3 in vivo. Collectively, our findings show that a substantially improved understanding of RNA-binding protein regulatory networks can be obtained through detailed analysis of motif quality.

## Results

### Deadenylation is stimulated by Puf3 and Zfs1 in vitro

To characterize substrate-selective deadenylation in the presence of RNA-binding proteins, we reconstituted this process using purified proteins. Full-length Zfs1 and Puf3 were expressed recombinantly as N-terminal maltose-binding protein (MBP) fusion proteins, and intact recombinant seven-subunit *S. pombe* Ccr4-Not was purified as described previously (*Stowell et al., 2016*). Pull-down assays revealed that Puf3 and Zfs1 interact with isolated Ccr4-Not (*Figure 1—figure supplement 1A*). Therefore, like metazoan homologs, *S. pombe* Puf3 and Zfs1 bind directly to Ccr4-Not.

By using a short model RNA substrate, deadenylation can be visualized at single-nucleotide resolution, and can be accurately quantified (*Webster et al., 2017*). To investigate whether Puf3 and Zfs1 stimulate the deadenylation activity of Ccr4-Not in vitro, we designed model RNA substrates containing the sequence motif recognized by either Puf3 or Zfs1 (PRE UGUAAAUA, or ARE UUAUUUAUU respectively, *Figure 1—figure supplement 1B*). These motifs were embedded within a synthetic 23-nucleotide sequence, upstream of a 30-adenosine poly(A) tail. Electrophoretic mobility shift assays confirmed that Puf3 and Zfs1 bind stably to their respective target RNAs (*Figure 1—figure supplement 1C*). The 1:1 stoichiometric protein-RNA complexes formed under these conditions were used as substrates in assays measuring the deadenylation activity of Ccr4-Not.

Ccr4-Not removed the poly(A) tails of both Puf3 and Zfs1 substrate RNAs in vitro. Puf3 and Zfs1 each stimulated deadenylation of their target RNAs: In the absence of Puf3, complete removal of the poly(A) tail from PRE-containing RNA occurred in approximately 32 min, whereas in the presence

of Puf3 the reaction was complete in less than 2 min (*Figure 1A* and *Figure 1—figure supplement 1D–E*). Therefore, addition of Puf3 increased the rate of deadenylation more than 20-fold. Zfs1 similarly increased the rate of deadenylation of its target RNA to less than 2 min (*Figure 1B* and *Figure 1—figure supplement 1D–E*). Thus, both Puf3 and Zfs1 have striking effects on the rate of deadenylation.

Non-adenosine residues upstream of the poly(A) tail were also rapidly removed by Ccr4-Not when Puf3 or Zfs1 were present. The exonucleases stop when they reach nucleotides that correspond to the binding site of the protein, suggesting that either the RNA-binding proteins cannot be released by Ccr4-Not, or that deadenylation becomes very slow after release of the RNA binding protein. This is consistent with a model in which the specificity of Ccr4-Not for adenosine is substantially less evident when it is tethered to the RNA substrate for a prolonged period of time (*Finoux and Séraphin, 2006*). In addition, the deadenylation reaction appeared to be more processive in the presence of Puf3 or Zfs1 because RNA lacking a poly(A) tail was visible when RNA with an intact tail was still present in the reaction (*Figure 1—figure supplement 1F*). This indicates that the RNA-binding protein increases the stability of the interaction of Ccr4-Not with the target RNA substrate during the course of the reaction.

## Ccr4-Not has an intrinsic RNA preference

In the absence of additional protein, Ccr4-Not is approximately 3-fold more active on the ARE-containing substrate than on the PRE-containing substrate (*Figure 1* and *Figure 1—figure supplement 1E*). It is generally believed that additional protein factors confer mRNA-selectivity to Ccr4-Not, but this finding indicates Ccr4-Not may possess a moderate degree of intrinsic mRNA selectivity. We previously observed that Ccr4-Not was less active on RNA substrates with secondary structure in the upstream 3'-UTR (*Stowell et al., 2016*). The free energies of the predicted most stable structures calculated with RNAfold software are −1.2 kcal/mol (Puf3-target RNA) and −0.2 kcal/mol (Zfs1-target RNA). Thus, neither of the substrates tested here is predicted to form stable RNA secondary structure. Our data suggest that the rate at which different mRNAs are deadenylated in vivo may be modulated by sequence-selectivity intrinsic to the Ccr4-Not complex.

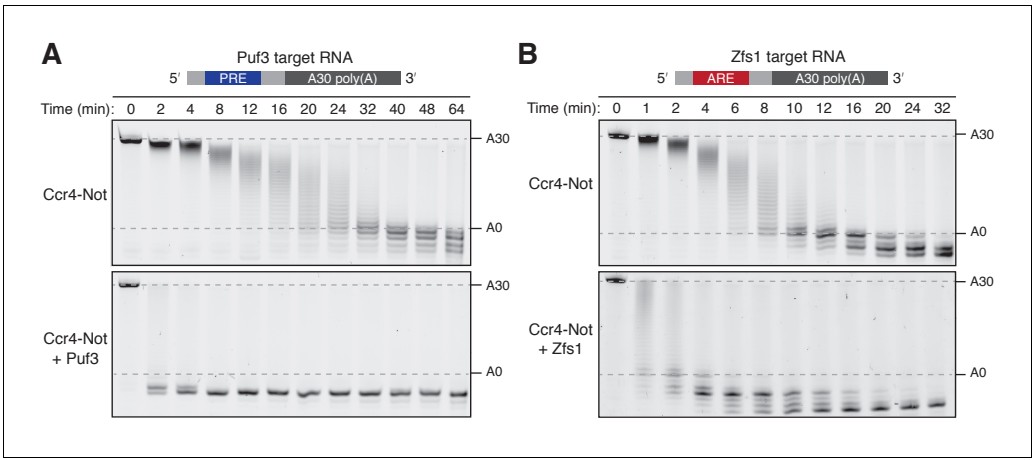

**Figure 1.** Puf3 and Zfs1 accelerate deadenylation by Ccr4-Not in vitro. Deadenylation by purified Ccr4-Not in the absence and presence of (**A**) Puf3 and (**B**) Zfs1. Synthetic RNA substrates were designed to contain a 30-adenosine poly(A) tail downstream of 23 non-poly(A) ribonucleotides including the recognition elements of Puf3 (PRE) or Zfs1 (ARE). RBP-RNA complexes were prepared at a 1:1 molar ratio before the addition of Ccr4-Not. Reactions were stopped at the indicated times and analyzed by denaturing PAGE. The sizes of the RNA substrate with and without the poly(A) tail are indicated by dashed lines.

DOI: https://doi.org/10.7554/eLife.40670.003

The following figure supplement is available for figure 1:

**Figure supplement 1.** Acceleration of deadenylation by Puf3 and Zfs1.
DOI: https://doi.org/10.7554/eLife.40670.004

## Puf3 and Zfs1 recruit Ccr4-Not via extended low-complexity sequences

As mentioned above, specific interactions between Ccr4-Not and Pumilio/TTP proteins had been previously described (*Fabian et al., 2013*; *Goldstrohm et al., 2006*; *Van Etten et al., 2012*). To dissect the mechanism of accelerated deadenylation and to determine the contributions of different parts of Puf3 and Zfs1, we made a series of mutations and truncations in these proteins, and tested their abilities to stimulate deadenylation. First, we tested whether the RNA-binding domains alone (Puf3$^{PUM}$ and Zfs1$^{TZF}$, *Figure 2—figure supplement 1A–B*) affected deadenylation. Addition of Puf3$^{PUM}$ caused a modest acceleration (complete deadenylation in ~20 min compared to ~32 min; *Figure 2A* and *Figure 2—figure supplement 1C,D*). This is consistent with a reported interaction between the Pumilio domain and the Caf1/CNOT7 subunit of Ccr4-Not, that tethers RNA directly to the nuclease subunit to accelerate deadenylation (*Goldstrohm et al., 2006*; *Van Etten et al., 2012*).

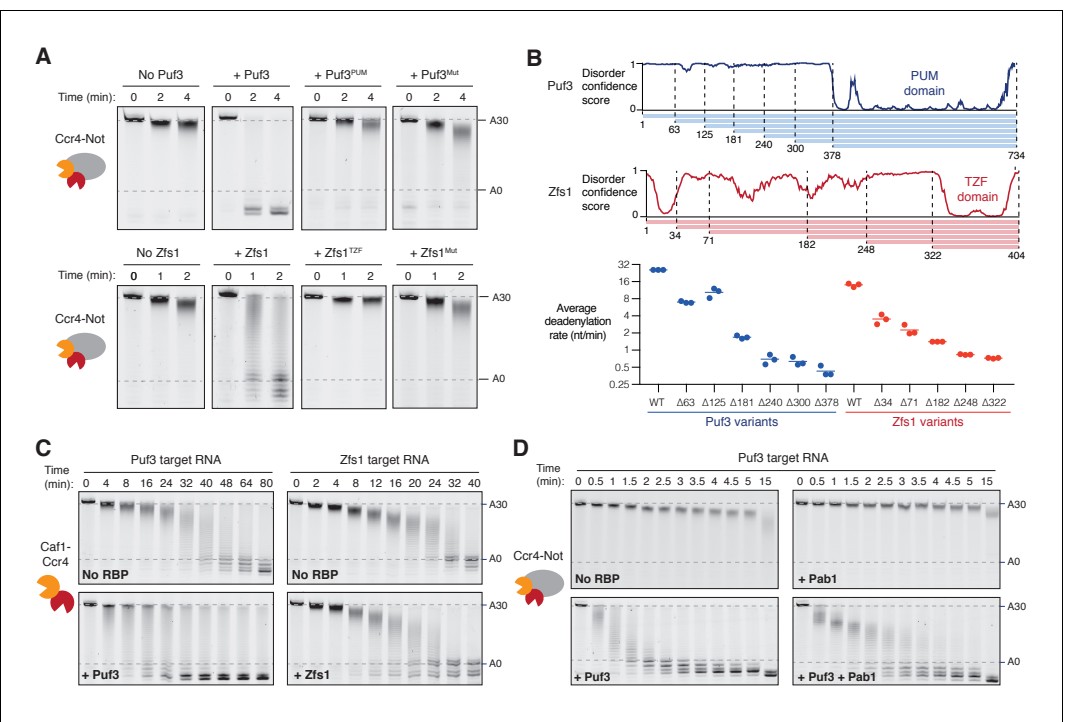

**Figure 2.** Recruitment of Ccr4-Not by Puf3 and Zfs1. Accelerated deadenylation by RNA binding proteins requires both a low-complexity region and RNA-binding activity. (**A**) Deadenylation by purified Ccr4-Not in the presence of the indicated RNA-binding protein variants was analyzed by denaturing PAGE. Puf3$^{PUM}$ and Zfs1$^{TZF}$ each comprise the RNA-binding domain alone; Puf3$^{Mut}$, Zfs1$^{Mut}$ are full-length proteins with point mutations that abolish RNA binding. Panels for no Puf3, no Zfs1, and wild-type proteins are reproduced from *Figure 1*. (**B**) The low-complexity domains of Puf3 and Zfs1 contain multiple regions that contribute to deadenylation by Ccr4-Not. The upper panel shows diagrams of Puf3 and Zfs1 with regions of predicted disorder and truncated construct lengths. The bottom panel shows deadenylation by purified Ccr4-Not in the presence of the indicated protein variants, analyzed by denaturing PAGE. The most abundant RNA size decreased linearly with time, and average reaction rates in number of adenosines removed/min were determined by linear regression on triplicate measurements. Each experiment is plotted as a point and the mean is shown as a horizontal line. Statistical significance is shown in *Figure 2—figure supplement 1G*. (**C**) Deadenylation by the Caf1-Ccr4 nuclease subcomplex in the absence (No RBP) and presence of Puf3 and Zfs1. (**D**) Puf3-stimulated deadenylation by Ccr4-Not in the absence or presence of Pab1 (1:1 molar ratio to RNA).

DOI: https://doi.org/10.7554/eLife.40670.005

The following figure supplements are available for figure 2:

**Figure supplement 1.** Characterization of Puf3 and Zfs1 domains required for accelerated deadenylation.

DOI: https://doi.org/10.7554/eLife.40670.006

**Figure supplement 2.** Puf3 accelerates deadenylation in the presence of Pab1.

DOI: https://doi.org/10.7554/eLife.40670.007

Since stimulation by Puf3[PUM] was much less than with full-length Puf3, regions outside the Pumilio domain likely play an important role in the recruitment of Ccr4-Not. Interestingly, Zfs1[TZF] was inhibitory to deadenylation (*Figure 2A* and *Figure 2—figure supplement 1C*). This may be because this domain conceals sequence elements that confer the higher level of intrinsic activity on the ARE-containing RNA.

To confirm that RNA binding is essential to the function of Puf3 and Zfs1, we generated full-length proteins with mutations in the RNA-binding interface (Puf3[Mut]: Y449A/Y671A and Zfs1[Mut]: F349A/F387A). Compared to the wild-type proteins, these mutants had a substantially reduced ability to bind their substrate RNAs, and to accelerate deadenylation (*Figure 2A* and *Figure 2—figure supplement 1B–E*). Together, these data show that the strong stimulatory effect of Puf3 and Zfs1 depends on RNA binding, but the RNA-binding domains alone are not sufficient.

In addition to a C-terminal RNA-binding domain, both Puf3 and Zfs1 contain an extended N-terminal region of low amino acid sequence complexity that is predicted to be predominantly disordered (*Figure 2B*). We hypothesized that short sequence motifs within the low-complexity regions of Puf3 and Zfs1 interact with Ccr4-Not, as has been demonstrated for other interaction partners (*Bhandari et al., 2014*; *Fabian et al., 2013*; *Raisch et al., 2016*). Thus, we generated a series of N-terminal truncations and measured their effects on the deadenylation activity of Ccr4-Not (*Figure 2B* and *Figure 2—figure supplement 1F–G*). For both proteins, no single truncation accounted for the stimulatory effect on deadenylation. Instead, as the length of low-complexity sequence was reduced, the stimulatory effect on the deadenylation rate was also reduced. It is therefore likely that multiple sequence elements, distributed throughout the low-complexity regions, interact with Ccr4-Not.

## Non-enzymatic subunits of Ccr4-Not contribute to the interaction with Puf3 and Zfs1

Two of the seven core subunits of Ccr4-Not are poly(A)-selective exonucleases, Caf1 and Ccr4. To determine whether Puf3 and Zfs1 can stimulate deadenylation in the absence of the non-enzymatic subunits, we performed reactions with a purified dimeric subcomplex comprised of Caf1 and Ccr4 (*Stowell et al., 2016*). Puf3 accelerated the rate of deadenylation by Caf1-Ccr4 substantially less than it accelerated intact Ccr4-Not (2-fold and >20 fold respectively, *Figure 2C* left compared to 1A). Fully-deadenylated and non-deadenylated RNA was observed simultaneously during both reactions, suggestive of a processive reaction. Furthermore, non-adenosine nucleotides were also readily removed by the Caf1-Ccr4 complex. This suggests that Puf3 interacts with Caf1-Ccr4, but association occurs more slowly when these subunits are not incorporated into the intact Ccr4-Not complex.

Zfs1 did not substantially affect the deadenylation activity of the Caf1-Ccr4 dimeric complex (*Figure 2C*, right). Thus, our data are consistent with Puf3 and Zfs1 recruiting the Ccr4-Not complex primarily through an interaction between their low-complexity regions and the non-enzymatic subunits of Ccr4-Not.

## Puf3 accelerates the release of Pab1

Poly(A)-binding protein (Pab1) contributes to translation initiation, and its release from mRNAs can repress translation (*Parker, 2012*). Ccr4-Not efficiently releases Pab1 from the poly(A) tail through removal of the nucleotides to which it is bound (*Webster et al., 2018*). It is not known, however, how Pab1 influences accelerated deadenylation. We therefore examined deadenylation of Pab1-bound RNA in the presence of Puf3.

Electrophoretic mobility shift assays showed that stoichiometric amounts of Puf3 and Pab1 could be simultaneously loaded onto the RNA substrate (*Figure 2—figure supplement 2A*). In deadenylation assays, the addition of Puf3 increased the rate at which the Pab1-bound poly(A) tail was removed by ~8 fold (*Figure 2D* and *Figure 2—figure supplement 2B*). Therefore, Puf3 can promote release of Pab1 by accelerating deadenylation by Ccr4-Not. Still, Pab1 slows the overall rate of Puf3-accelerated deadenylation, suggesting that the rate of Pab1 release limits the very fast rate of deadenylation that occurs when Ccr4-Not is recruited by Puf3.

We previously reported that Caf1 and Ccr4 have similar activities within the purified Ccr4-Not complex (*Stowell et al., 2016*). More recently, we showed that the presence of Pab1 differentiates the activity of Ccr4 and Caf1: only Ccr4 removes adenosines bound by Pab1 (*Webster et al., 2018*).

It is not known, however, whether Caf1 and Ccr4 are differentially active when RNA-binding proteins recruit Ccr4-Not to specific mRNAs. We tested the contribution of Ccr4 and Caf1 to Puf3-accelerated deadenylation using purified Ccr4-Not variants with point mutants that abolish the catalytic activity of either nuclease.

Using our in vitro assays, Puf3 accelerated deadenylation by both Ccr4-inactive and Caf1-inactive complexes (*Figure 2—figure supplement 2C*). This indicates that deadenylation in the presence of Puf3 is mediated by either nuclease. In the presence of Puf3 and Pab1, Ccr4-inactive Ccr4-Not did not proceed beyond ~20 adenosines (*Figure 2—figure supplement 2C*). This is similar to when Puf3 was not present (*Webster et al., 2018*). Thus, Pab1, and not Puf3, differentiates the two nucleases of Ccr4-Not.

## Puf3 and Zfs1 induce substrate-selective deadenylation by Ccr4-Not

Having established that Puf3 and Zfs1 are capable of accelerating Ccr4-Not activity, we sought to test whether they are also sufficient for substrate-selectivity. We began by examining whether the presence of the target motif recognized by each protein was essential for accelerated deadenylation. The effect of Puf3 and Zfs1 on its own target RNA was compared with its effect on a non-target RNA. For the latter, we used the RNA target of the other protein (Puf3 with Zfs1-target RNA, and Zfs1 with Puf3-target RNA). We were surprised to find that each protein stimulated deadenylation of an RNA that did not contain its target motif (*Figure 3A*). Quantification of deadenylation rates showed that on-target deadenylation was approximately 3–4 fold faster than off-target deadenylation, which was in turn 2–5 fold faster than when no additional protein was present (*Figure 3—figure supplement 1A*). Hence, the correct target RNA motif contributes to the ability of an RNA-binding protein to accelerate deadenylation in vitro, but is not absolutely required.

Since Puf3 with a mutated RNA-binding domain still stimulates deadenylation, allosteric activation of Ccr4-Not by the Puf3 low complexity region may contribute to this accelerated deadenylation of off-target RNAs (*Figure 2—figure supplement 1D*). However, mutant Zfs1 that does not bind RNA doesn't accelerate deadenylation. Thus, allosteric activation is not likely to be a general mechanism. Instead, we hypothesized that off-target stimulation of deadenylation in our in vitro experiments is due to off-target RNA binding.

To determine the binding affinities of full-length Puf3 and Zfs1 for the RNA substrates, we performed fluorescence polarization RNA binding assays with RNAs containing the target motif of Puf3 or Zfs1, as well as 30-mer polyadenosine (*Figure 3B*). Each RNA-binding protein bound to its respective target RNA sequence with high affinity ($K_D$ = 1–2 nM). In contrast, Puf3 bound to the off-target ARE sequence with $K_D$ of 120 nM, while Zfs1 bound the off-target PRE sequence with $K_D$ of 850 nM. In both cases, poly(A) was bound with ~100 fold lower affinity than the target RNA. Hence, both Puf3 and Zfs1 are selective in RNA binding, with >100 fold higher affinity for their target RNA than off-target RNAs. Still, Puf3 and Zfs1 bind off-target sequences with nanomolar affinities and this could account for their abilities to stimulate deadenylation of off-target RNAs.

We hypothesized that Puf3 and Zfs1 would select their targets from a mixture of target and non-target RNAs to promote targeted deadenylation. To test this, we developed a system to directly examine selective deadenylation by Ccr4-Not. Two polyadenylated RNA substrates labelled with different fluorophores were combined in stoichiometric quantities before the addition of Ccr4-Not and analysis of deadenylation activity. The same gel could therefore be used to monitor deadenylation of both the fluorescein-labelled Puf3-target substrate, and the Alexa647-labelled Zfs1-target substrate, displayed in blue and red respectively (*Figure 3C*). In this two-color assay, the Zfs1-target RNA was deadenylated by Ccr4-Not 3-fold faster than the Puf3-target RNA in the absence of additional protein. This is consistent with intrinsic selectivity of Ccr4-Not that was observed when each RNA was examined in isolation (*Figure 1*).

We then performed a series of reactions using three different concentrations of Puf3 or Zfs1, and each RNA at 100 nM. With 50 nM Puf3 or Zfs1, deadenylation was accelerated only on the target RNA (*Figure 3D*, top; blue for Puf3-target and red for Zfs1-target). A fraction of the target RNA was unaffected, consistent with incomplete occupancy of the RNA-binding protein on its target. With 250 nM Puf3 or Zfs1, deadenylation of all target RNA was substantially increased while deadenylation of the non-target RNA was unchanged (*Figure 3D*, middle). Thus, Puf3 and Zfs1 promote selective, accelerated deadenylation. Selectivity was lost when an excess of the RNA-binding protein was added (1000 nM) and all RNA was deadenylated rapidly (*Figure 3D*, bottom).

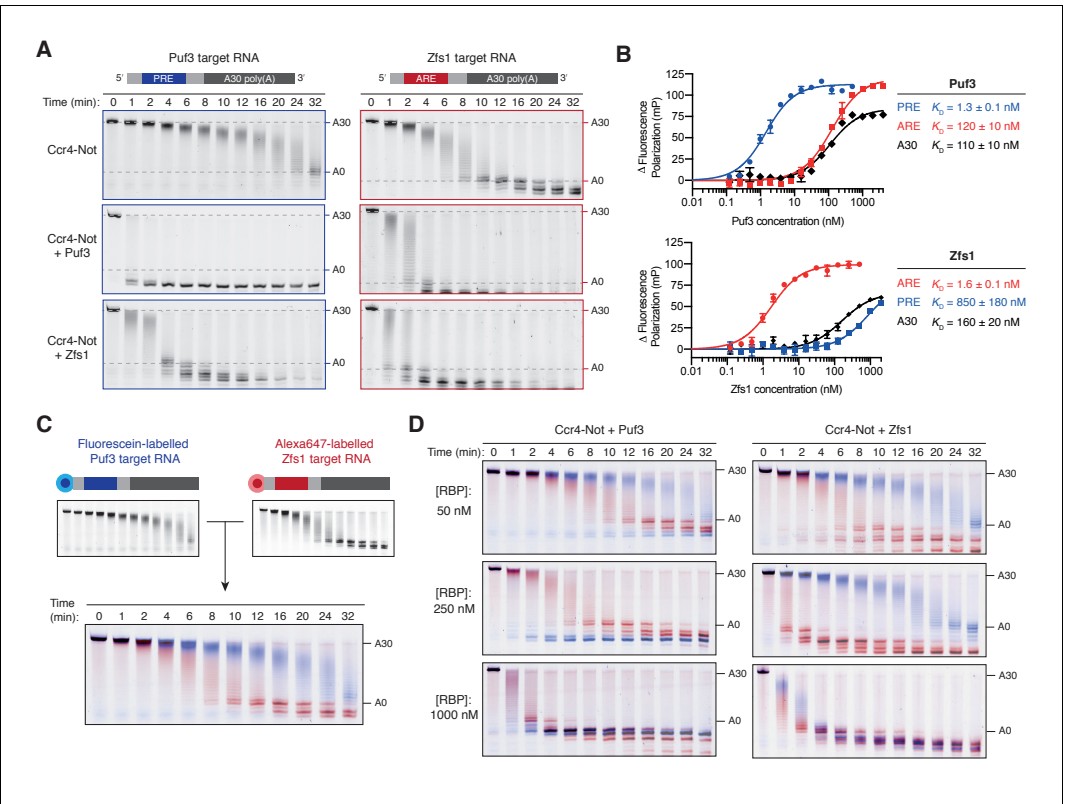

**Figure 3.** Puf3 and Zfs1 mediate selective deadenylation by Ccr4-Not. (**A**) Puf3 and Zfs1 accelerate deadenylation of RNAs with or without a target recognition motif. Puf3 or Zfs1 was added to PRE-containing RNA (Puf3-target) or ARE-containing RNA (Zfs1-target) before addition of Ccr4-Not. Deadenylation was analyzed by denaturing PAGE. Top right and bottom right panels are reproduced from *Figure 1B*. (**B**) Fluorescence polarization experiments assaying the binding of full-length Puf3 (top) or Zfs1 (bottom) to 5'−6FAM labelled RNAs. Error bars represent the standard error in triplicate measurements. (**C**) Deadenylation of a mixture of 100 nM fluorescein-labelled Puf3-target RNA (blue) and 100 nM Alexa647-labelled Zfs1-target RNA (red) analyzed by denaturing PAGE. Gels were scanned to detect each fluorophore before signals were overlaid. (**D**) Puf3 and Zfs1 promote selective deadenylation. Reactions were performed as in (**C**) but with the addition of Puf3 (left) or Zfs1 (right) at the indicated concentrations.

DOI: https://doi.org/10.7554/eLife.40670.008

The following figure supplement is available for figure 3:

**Figure supplement 1.** Analysis of RNA selectivity during deadenylation.

DOI: https://doi.org/10.7554/eLife.40670.009

To examine RNA binding by Puf3 and Zfs1 under the same conditions, we performed electrophoretic mobility shift assays. Consistent with deadenylation assays, target RNA binding is complete and selective at 250 nM, but is non-selective at 1000 nM (*Figure 3—figure supplement 1B*). In conclusion, Puf3 and Zfs1 are sufficient for substrate-selective deadenylation by Ccr4-Not, within a given concentration window. At higher concentrations, they also bound to and accelerated deadenylation on RNAs that do not contain their target motif.

## Puf3 distinguishes between very similar motifs

In the cell, RNAs with very similar sequences are present but RNA-binding proteins distinguish between these to act preferentially on specific targets. Yeasts possess multiple Pumilio proteins, and these recognize RNA motifs similar in sequence (*Wickens et al., 2002*). For example, an RNA target of *S. cerevisiae* (*Sc*) Puf3 differs from a target of *Sc*Puf5/Mpt5 in two nucleotide positions within a core 8-nt motif (*Goldstrohm et al., 2006*) (*Figure 4A*, top). Given that Puf3 binds off-target RNAs at

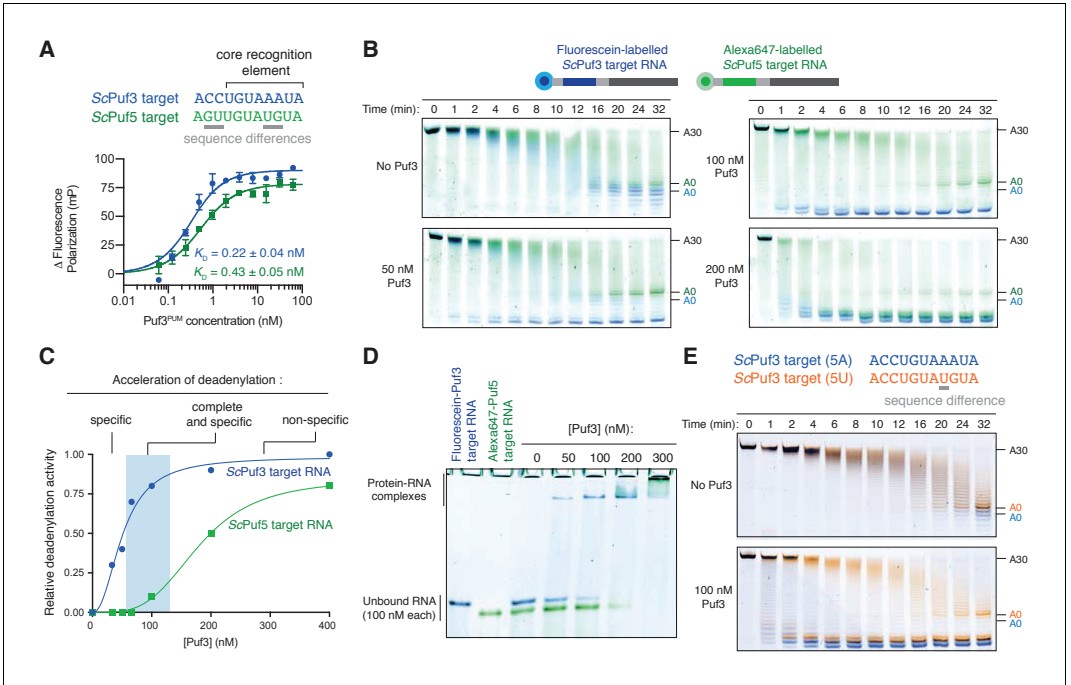

**Figure 4.** Puf3 distinguishes between similar RNA motifs. (**A**) Fluorescence polarization experiments assaying the binding of Puf3 RNA-binding domain to RNA containing a *Sc*Puf3 motif (blue) or a *Sc*Puf5 motif (green). Error bars represent the standard error in triplicate measurements. (**B**) Deadenylation of a mixture of fluorescein-labelled *Sc*Puf3-target RNA (blue) and Cy3-labelled *Sc*Puf5-target RNA (green) analyzed by denaturing PAGE. Gels were scanned to detect each fluorophore before signals were overlaid. (**C**) Selective deadenylation occurs within a narrow range of Puf3 concentrations. Plot of relative deadenylation rates of *Sc*Puf3-target RNA and *Sc*Puf5-target RNA in reactions as in (**B**) at a series of Puf3 concentrations. Reactions were quantified based on the fraction of RNA fully deadenylated within 8 min. (**D**) Electrophoretic mobility shift assay showing the binding of Puf3 to a mixture of fluorescein-labelled *Sc*Puf3-target RNA (blue) and Cy3-labelled *Sc*Puf5-target RNA (green). The concentration of each RNA was 100 nM. (**E**) Deadenylation of a mixture of fluorescein-labelled *Sc*Puf3(5A)-target RNA (blue) and Alexa647-labelled *Sc*Puf3(5U)-target RNA (orange) analyzed by denaturing PAGE. Gels were scanned to detect each fluorophore before signals were overlaid.

DOI: https://doi.org/10.7554/eLife.40670.010

higher concentrations (***Figure 3—figure supplement 1B***), we investigated whether the fission yeast Puf3 distinguishes between the very similar motifs of *Sc*Puf3 and *Sc*Puf5 in vitro.

We used fluorescence polarization experiments to examine RNA binding and found that *S. pombe* Puf3[PUM] binds to both the *Sc*Puf3 target and the *Sc*Puf5 target with high affinity (***Figure 4A***, bottom). The affinity for the *Sc*Puf3 motif is ~2 fold higher than for the *Sc*Puf5 motif, but the binding affinity in both cases is in the picomolar range. Surprisingly, in deadenylation activity assays with 100 nM Puf3, we found that the *Sc*Puf3 target was selectively deadenylated in preference to the *Sc*Puf5 target (***Figure 4B***). Thus, Puf3 can distinguish between very similar RNAs and selectively target them for deadenylation by Ccr4-Not.

Selective deadenylation only occurred within a narrow range of Puf3 concentrations where the quantity of Puf3 protein is less than or equal to the quantity of *Sc*Puf3-target RNA (***Figure 4B,C***). With less Puf3 (50 nM Puf3), not all *Sc*Puf3-target RNA was rapidly deadenylated. Conversely, when a molar excess of Puf3 was present (200 nM Puf3), both the *Sc*Puf3- and *Sc*Puf5-target RNAs were deadenylated rapidly. We performed RNA EMSAs and found that the ability of Puf3 to stimulate deadenylation correlated with RNA binding: binding to *Sc*Puf5-target RNA occurred only when Puf3 protein was in molar excess to the *Sc*Puf3-target RNA (***Figure 4D***). Complete and selective binding by Puf3, and selective deadenylation by Ccr4-Not, therefore only occurs within a narrow window of Puf3 protein concentration.

We next examined whether Puf3 can distinguish between sequences that are even more similar. Within the 8-nucleotide motif recognized by ScPuf3, the fifth position is the most variable, and A or U are thought to be tolerated (*Kershaw et al., 2015*). This is a property shared by the human homologs PUM1 and PUM2 (*Galgano et al., 2008*; *Morris et al., 2008*). In the previous experiments, the RNA motif contained A at position 5 (now called ScPuf3-target(5A)). We tested whether there is a preference for A or U at position five using the two-color deadenylation assay and found that the ScPuf3-target(5A) RNA was deadenylated in preference to ScPuf3-target(5U) RNA (*Figure 4E*). Thus, Puf3 has a remarkable capacity to distinguish between sequences within the current definition of its target motif. The identity of the nucleotide in position five likely affects 'motif quality', and Puf3 interacts with sequences of different quality in a priority-based manner.

## Kinetic basis of selective RNA binding by Puf3

To gain insight into the selective RNA binding by Puf3 that allows it to distinguish between RNAs of very similar sequences, we analyzed the binding kinetics using SwitchSENSE (*Knezevic et al., 2012*). Switchable nanolevers were functionalized with RNAs containing each of the three RNA motifs tested in the deadenylation reactions: ScPuf3-target(5A), ScPuf3-target(5U), and ScPuf5-target. We measured the rates of association ($k_{on}$) and dissociation ($k_{off}$) of fission yeast Puf3[PUM] binding to each of these, as well as to polyadenosine as a negative control.

Rates of association followed the same pattern of priority binding that we had identified by deadenylation assays and EMSA (*Figure 5A*): Association with the ScPuf3-target(5A) was ~25% faster than with ScPuf3-target(5U) and ~three fold faster than with the ScPuf5-target. Association with polyadenosine was even slower (~seven fold). Thus, RNA sequence affects the rate of Puf3 association.

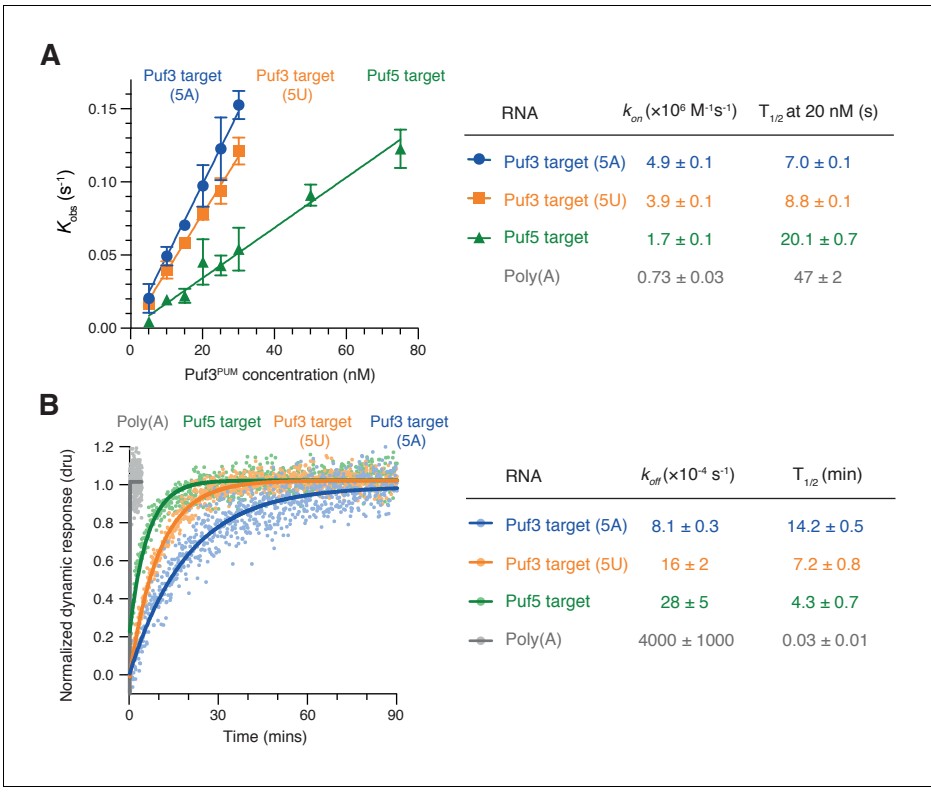

**Figure 5.** RNA motif quality affects Puf3 binding kinetics. (**A**) The kinetic rates of association ($k_{on}$) of purified Puf3[PUM] with RNA containing the indicated *S.cerevisiae* sequence motifs were determined by SwitchSENSE. Error bars are the standard error. (**B**) Representative SwitchSENSE sensograms showing the dissociation of Puf3[PUM] from the indicated *S. cerevisiae* RNA sequences. dru, dynamic response units. In (**A**) and (**B**), rate constants and half-lives are shown with standard error for measurements performed in triplicate.
DOI: https://doi.org/10.7554/eLife.40670.011

Importantly, however, all sequences were bound rapidly: at a concentration of 20 nM, binding to all RNAs took place on the timescale of seconds (*Figure 5A*, right).

When we examined rates of dissociation, we found that Puf3 remained bound to the *Sc*Puf3-target with A at position five for twice as long as it remained bound to the RNA with U at position 5, and ~three times as long as it was bound to the *Sc*Puf5-target RNA (*Figure 5B*). Dissociation from polyadenosine occurred more than 100-fold faster. Thus, rates of Puf3 dissociation also reflected the preference order that we had observed in deadenylation experiments. Combining our association and dissociation measurements, sequence-selective binding by Puf3 is expected to occur by the combination of rapid sampling of all RNAs with slower dissociation from sequences with higher motif quality.

## Motif quality affects *Sc*Puf3-bound transcripts in vivo

The ability of Puf3 to discriminate between similar sequences is likely to play a role in defining the mRNAs within its regulatory network. To examine the contribution of RNA motif quality to Puf3 binding in vivo, we analyzed several published studies of *S. cerevisiae* Puf3. The mRNAs that contain Puf3 motifs are different between species, but we expected motif quality to be similarly important in all organisms (*Hogan et al., 2015*). Thus, we applied our in vitro findings using *S. pombe* proteins to a computational analysis of *S. cerevisiae* Puf3.

We compiled a list of 1331 putative *Sc*Puf3 mRNA targets from three published sources. The first study of transcriptome-wide *Sc*Puf3 binding identified 220 targets by RNA immunoprecipitation followed by microarray analysis (RIP-chip) (*Gerber et al., 2004*). A more recent study using RNA immunoprecipitation followed by sequencing (RIP-seq) identified 1132 significantly enriched transcripts (*Kershaw et al., 2015*): the larger number of targets identified was attributed to the improved sensitivity of deep sequencing. A third study detected in vivo binding of *Sc*Puf3 to 476 mRNAs using an RNA-tagging method (*Lapointe et al., 2015*). While there is substantial overlap between the binding targets from each list, 74% of mRNAs were identified in only one of the three studies (*Figure 6A*). The set of mRNAs that are direct targets of *Sc*Puf3 regulation therefore remains unclear.

We began by computationally identifying the most enriched sequence motif in the complete set of 1331 mRNAs using MEME (*Bailey et al., 2015*). This had seven conserved nucleotides within an 8-nt motif (UGUANAUA), and was present in 627 mRNAs (41% of total) (*Figure 6B*, top). The nucleotide at position five is variable, consistent with previous analyses of transcriptome-wide data (*Gerber et al., 2004*; *Kershaw et al., 2015*).

The best-characterized *Sc*Puf3 binding site is within the *COX17* mRNA (*Olivas and Parker, 2000*). This mRNA was also the most highly enriched following RNA-immunoprecipitation (*Kershaw et al., 2015*). Interestingly, it has sequence features outside the seven core nucleotides that contribute to the particularly high affinity binding of *Sc*Puf3: cytosine at position –2 and adenosine at position +5 (*Zhu et al., 2009*). Having identified that fission yeast Puf3 has the capacity to distinguish between similar sequences, we hypothesized that these additional sequence features might provide a better definition of *Sc*Puf3 targets transcriptome-wide. Thus, we categorized target mRNAs by the presence of a *Sc*Puf3-target motif as well as the nucleotide identity at positions –two and +5. We ranked these four groups by motif quality based on previously measured affinity values (*Figure 6B* and *Supplementary file 1*) (*Zhu et al., 2009*). Many of the predicted *Sc*Puf3-target transcripts that do not fit into one of these four groups do not contain any of these sequences but have a motif in which one of the core seven nucleotides are altered (denoted here as 'Δcore').

mRNAs that increase in steady-state abundance upon deletion of *PUF3* represent high-confidence targets of *Sc*Puf3-mediated mRNA decay. 82 such mRNAs were identified previously (*Kershaw et al., 2015*), and we found that motif 1, 2 or 3 is present in 73 of these (89%) (*Figure 6C* and *Supplementary file 1*). Furthermore, in 71 of the mRNAs that are increased in abundance upon *PUF3* deletion, the motif is located within the 3' UTR. Hence, the quality and the location of *Sc*Puf3 motifs both contribute to mRNA-destabilization.

We examined whether motif quality is correlated with the amount of *Sc*Puf3 bound to mRNAs in vivo and found that transcripts that were most enriched in *Sc*Puf3 immunoprecipitation experiments (*Kershaw et al., 2015*) had motif 1, 2 or 3 located in their 3'-UTR (*Figure 6D* and *Supplementary file 1*). In contrast, transcripts containing a group 4 motif, or a mismatch within the core seven nucleotides, generally had less *Sc*Puf3 bound. Interestingly, mRNAs with high quality

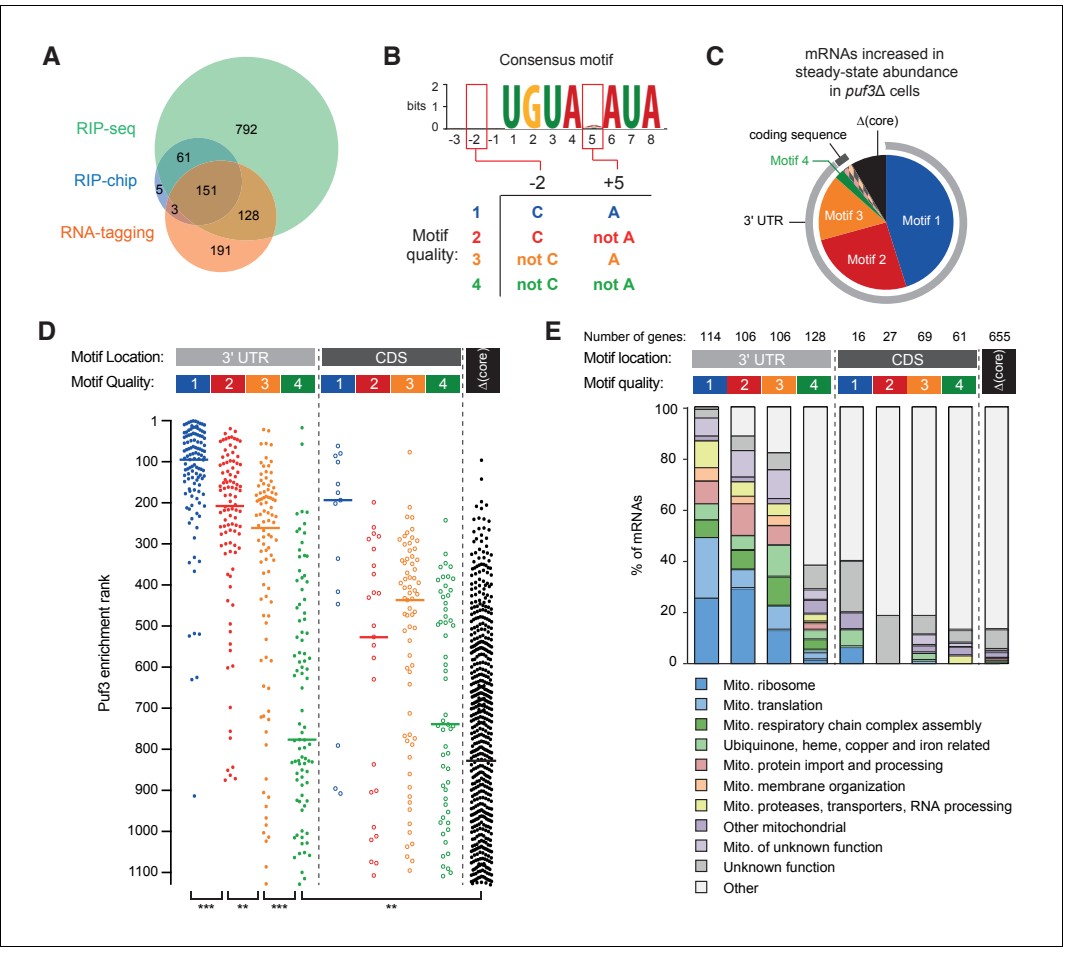

**Figure 6.** RNA motif quality defines a coherent mRNA regulatory network. (**A**) Overlap between the putative mRNAs bound by *S.cerevisiae* Puf3 determined by RIP-seq (*Kershaw et al., 2015*), RIP-chip (*Gerber et al., 2004*) and RNA-tagging (*Lapointe et al., 2015*). The numbers of mRNAs identified in each study is given. (**B**) Consensus motif identified from the complete set of putative *Sc*Puf3 mRNA targets. Definition of categories of Puf3 motif quality is shown below. (**C**) Diagram showing RNA motif quality and location in mRNAs that increase in abundance upon *PUF3* deletion (*Kershaw et al., 2015*). (**D**) *Sc*Puf3 binding versus RNA motif quality and location. Median enrichment rank is indicated with horizontal line. Statistical significance was evaluated with a Mann-Whitney rank comparison test. **, p=0.01; ***, p=0.001. (**E**) mRNAs with high quality *Sc*Puf3-target motifs encode proteins related to mitochondrial biogenesis.

DOI: https://doi.org/10.7554/eLife.40670.012

The following figure supplement is available for figure 6:

**Figure supplement 1.** Evaluation of mRNA target prediction by motif quality analysis.

DOI: https://doi.org/10.7554/eLife.40670.013

motifs located within the coding sequence were detected but were consistently bound by less *Sc*Puf3. Whether this reflects a lower occupancy of *Sc*Puf3, or a reduced stability of the interaction, is unknown. In summary, RNA sequence is an excellent predictor of *Sc*Puf3 binding in vivo.

## Motif quality defines a functionally homogeneous regulatory network

In *S. cerevisiae*, Puf3 is a regulator of mitochondrial function (*Lee and Tu, 2015*; *Saint-Georges et al., 2008*). Consistent with this, targets of *Sc*Puf3 include mRNAs that encode proteins localized to the mitochondria and factors involved in the oxidative phosphorylation pathway (*Gerber et al., 2004*; *Kershaw et al., 2015*; *Lapointe et al., 2015*; *Lapointe et al., 2018*). Many transcripts identified as targets, however, show no clear link to this role (*Kershaw et al., 2015*). An

important question therefore remains whether the role of *Sc*Puf3 is limited to mitochondrial regulation.

We examined the link between the quality of *Sc*Puf3-target motifs in mRNAs and the functions of the proteins they encode. A more detailed examination of gene function revealed that of the functionally annotated mRNAs containing motifs 1, 2 or 3 in their 3′ UTR, 245 of 278 (88%) were related to mitochondrial biogenesis (*Figure 6E* and *Supplementary file 2*). This includes 74 of the 80 subunits of the mitochondrial ribosome (*Bieri et al., 2018*), but none of the cytoplasmic ribosome. Furthermore, the 14 mitochondrion-specific tRNA aminoacylation enzymes, but none of the five that are both mitochondrial and cytoplasmic, have a high quality *Sc*Puf3 motif. Interestingly, transcripts encoding proteins that assemble the respiratory chain complexes, but not nuclear-encoded subunits of the complexes subunits themselves, are *Sc*Puf3 targets. Most components of the mitochondrial protein import complexes TIM22, TIM23 and TOM; mitochondrial chaperones; and membrane insertion factors are also targets. In contrast, mRNAs containing group four motif or Δcore were not generally related to mitochondrial function.

Compared to previous lists of *Sc*Puf3 target candidates, analysis of motif quality yields a list of mRNA targets that is shorter and more functionally coherent (*Figure 6—figure supplement 1A*). To assess the limitations of predicting RNA-binding protein targets directly from mRNA sequence motif we examined *Sc*Puf3 binding and the sequence features of transcripts that had motif 1, 2 or 3 in the 3′-UTR but encoded proteins not evidently related to mitochondrial biogenesis (33 in total), as well as mRNAs related to mitochondrial biogenesis but with motif 4 in the 3′-UTR (33 transcripts). These represent possible false-positives and false-negative predicted targets respectively. mRNAs with motif 1, 2 or 3 that were not related to mitochondrial biogenesis had significantly less *Sc*Puf3 association than those that were related to mitochondria (*Figure 6—figure supplement 1B*). This suggests that despite the presence of a high-quality motif, *Sc*Puf3 binding is less enriched. Similarly, mRNAs with motif 4 that were related to mitochondrial biogenesis had higher levels of *Sc*Puf3, suggesting that despite sequences with lower motif quality, *Sc*Puf3 is selectively bound. Analysis of sequences surrounding the *Sc*Puf3-target motif revealed no consistent differences between functional groups (*Figure 6—figure supplement 1C*). Therefore, factors other than motif quality also influence *Sc*Puf3 binding and this is correlated with encoded function.

## Discussion

The recruitment of enzymatic multi-protein complexes by RNA-binding adaptor proteins often facilitates post-transcriptional regulation of gene expression (*Rissland, 2017*). Here, we show by biochemical reconstitution that Puf3 and Zfs1 function as molecular tethers during deadenylation, bringing together specific mRNAs and exonucleases (*Figure 7A*). Interactions between the low-complexity regions of Puf3/Zfs1 and non-enzymatic subunits of Ccr4-Not are critical to this process. Sequence-selectivity of the RNA-binding proteins largely determines the selection of target RNAs for deadenylation. Moreover, the stability of the interaction between Puf3 and RNA is exquisitely sensitive to the RNA sequence. In vivo, this plays a major role in defining the functionally coherent RNA regulatory network of Puf3 in budding yeast, and likely in other organisms.

### Ccr4-Not recruitment by RNA-binding proteins

RNA-binding proteins that interact with the mRNA decay machinery typically contain a canonical RNA-binding domain, and regions of low sequence complexity that are predicted to be structurally disordered (*Jonas and Izaurralde, 2013*). Short linear motifs within these regions can adopt secondary structure, and this may facilitate interaction with other proteins, including Ccr4-Not (*Bhandari et al., 2014*; *Fabian et al., 2013*; *Raisch et al., 2016*; *Sgromo et al., 2017*; *Stowell et al., 2016*). Recent work showed that *Drosophila* Nanos utilizes two short linear motifs to contact the NOT1 and NOT3 subunits of Ccr4-Not (*Raisch et al., 2016*). Likewise, the N- and C-terminal regions of human TTP interact with the CNOT1 and the CNOT9 subunits respectively (*Bulbrook et al., 2018*; *Fabian et al., 2013*; *Lykke-Andersen and Wagner, 2005*). We show that multiple regions within fission yeast Puf3 and Zfs1 contribute to Ccr4-Not recruitment, and these are distributed throughout the 300–400 amino acid region that is predicted to be predominantly disordered. Thus, the multi-partite mode of interaction for Puf3 and Zfs1 may be a general feature of RNA-binding proteins that interact with Ccr4-Not. The in vitro deadenylation assays described here

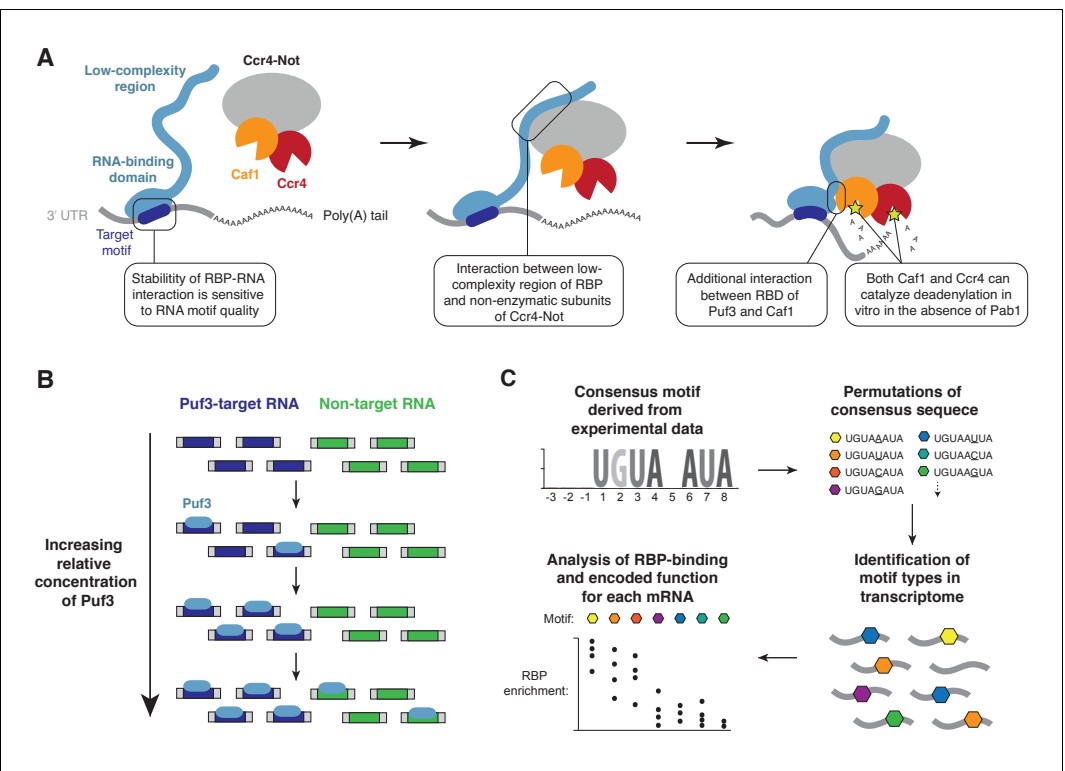

**Figure 7.** Models for substrate selectivity in deadenylation. (**A**) RNA-binding proteins (RBPs) promote targeted deadenylation by simultaneously binding to RNA via an RNA-binding domain (RBD) and to Ccr4-Not via a low-complexity region. These interactions can be regulated by post-translational modifications. (**B**) Model for priority-based binding of Puf3 to RNA. (**C**) Scheme for defining target sequences of RNA-binding proteins.
DOI: https://doi.org/10.7554/eLife.40670.014

represent a powerful tool for testing the importance of particular sequence regions in interaction partners and evaluating their redundancy.

The Pumilio family of RNA-binding proteins appeared to represent an exception to the pattern that Ccr4-Not recruitment occurs exclusively via short linear motifs within intrinsically disordered regions. The RNA-binding Pumilio domain of human PUM1 and *S. cerevisiae* Puf4 and Puf5 interacts directly with Caf1/CNOT8 (*Goldstrohm et al., 2006*). Consistent with this, in our study the *S. pombe* Puf3 PUM domain stimulated the rate of deadenylation by Ccr4-Not (*Figure 2—figure supplement 1C*), and full-length Puf3 stimulated deadenylation by a Caf1-Ccr4 subcomplex (*Figure 2C*). Importantly, however, both of these effects were weak relative to the stimulation of intact Ccr4-Not by full-length Puf3 (*Figure 1A*). Thus, the low-complexity region of *S. pombe* Puf3 and the non-enzymatic subunits of Ccr4-Not play important roles in the interaction. The Pumilio proteins are therefore likely similar to other Ccr4-Not interaction partners.

Intrinsically disordered regions are common sites of post-translational modification such as phosphorylation. TTP is a phosphorylation target in the p38 MAPK pathway, and this disrupts the recruitment of Ccr4-Not (*Marchese et al., 2010*). Similarly, phosphorylation occurs throughout the low-complexity region of *S. cerevisiae* Puf3, and this stops the rapid decay of transcripts to which it is bound (*Lee and Tu, 2015*). Given that the low-complexity region is involved in the recruitment of Ccr4-Not, it is likely that phosphorylation regulates this interaction. Rates of mRNA decay may be tuned by the extent of phosphorylation within the extended interface that contacts Ccr4-Not.

## A priority model of selective RNA binding

The correct recognition of mRNA targets by RNA-binding proteins is critical for recruitment of effector complexes, such as Ccr4-Not. We found that Puf3 and Zfs1 stimulate deadenylation of off-target RNAs, but they can discriminate between mixtures of RNA substrates to select their target. Titration of Puf3 under conditions where two similar RNA sequences were present showed that binding of RNA occurs by a model of priority: Puf3 binds to an optimal target when it is available, but after saturation of all available sites, it will readily associate with a sub-optimal target (*Figure 7B*). This model of RNA target selection, where an RNA-binding protein has a distribution of affinities for different RNA sequences, challenges the model where there is a sharp distinction between affinities for specific and non-specific RNAs (*Jankowsky and Harris, 2015*). The relative abundance of RNA-binding proteins and RNA therefore determines occupancy, and is integral to the definition of the target motif.

We propose that tight control of RNA-binding protein abundance in the cell is essential for maintenance of a stable set of target mRNAs. Interestingly, the mRNA encoding Puf3 in *S. cerevisiae* is itself a target of Puf3 (*Supplementary file 1*). This predicts a potential feedback mechanism by which *Sc*Puf3 protein abundance is auto-regulated. Furthermore, the *Sc*Puf3-recognition motif within the *Sc*Puf3 mRNA is within group 4 (*Figure 6B*) and this is at the boundary that separates targets from non-targets. The *Sc*Puf3 mRNA may only become destabilized when *Sc*Puf3 protein abundance becomes high enough that it begins to destabilize mRNAs containing motifs of lower quality that are not related to mitochondrial biogenesis.

Increasing the level of human PUM1 by overexpression causes a number of mRNAs to be repressed, indicative of off-target RNA binding, and PUM1 availability may be regulated by a long noncoding RNA (*Lee et al., 2016*; *Tichon et al., 2018*). In mice, PUM1 is expressed constitutively, but heterozygotes that contain only a single gene copy display reduced PUM1 protein levels (*Gennarino et al., 2015*). This haploinsufficiency produces severely impaired neuronal function that is due, at least in part, to impaired destabilization of the *ATAXIN1* mRNA. Thus, consistent with our model of RNA binding, the dosage of RNA-binding proteins is also important to maintain on-target mRNA expression.

## Identification of mRNA targets of RNA-binding proteins

Even when the consensus motif recognized by an RNA-binding protein is known, it is often difficult to predict which transcripts are targeted by that protein in vivo: A high affinity interaction is not sufficient for the sequence to be a preferred target, as is exemplified by the tight binding of fission yeast Puf3 to the RNA motif recognized by *Sc*Puf5 (*Figure 4A*). Our kinetic analysis of Puf3 binding to RNA revealed a fast rate of association with RNA that does not contain the optimal sequence motif. We conclude that Puf3-RNA dissociation rates are likely major contributors to the rates of deadenylation.

In vivo, Puf3 likely samples numerous mRNAs before remaining stably associated with an mRNA containing an optimal sequence motif. An important consequence of this is that RNA-immunoprecipitation (and especially use of covalent crosslinking) will detect mRNAs that are not genuine targets. An indication of in vivo dissociation rates of *Sc*Puf3 were previously obtained using RNA tagging (*Lapointe et al., 2015*). In this method, an enzyme that adds uridines to the 3′ end of RNA is attached to the RNA-binding protein of interest. The length of the U-tail on an mRNA correlated with whether the mRNAs are regulated by *Sc*Puf3 in the cell and is also expected to correlate with the duration the RNA-binding protein is associated with it (*Lapointe et al., 2015*). This study differentiated between Puf3 'sampling' of RNAs where the U-tails were short and the off-rate was too fast to allow regulation (deadenylation), and Puf3 'regulation' where the U-tails were longer, the off-rate was slower, and the interaction leads to a productive output (deadenylation). In addition to cellular concentration of RNA binding proteins and their targets, competition among RNA binding proteins with similar sequence preferences will also influence this kinetic model for productive interaction (*Lapointe et al., 2017*). Taken together, there is a striking convergence between the results of in vitro (this study) and in vivo (*Lapointe et al., 2015*) studies. Our analysis also correlates well with a more recent study that combined multiple methods to identify 91 Puf3 targets (*Lapointe et al., 2018*).

A major limitation in all experimental approaches is that the threshold separating genuine from non-genuine targets is difficult to define. Our analysis of the mRNA targets of *S. cerevisiae* Puf3 demonstrates, however, that accurate prediction may be possible from examination of the 3'-UTR sequence. If, as our data suggests, the function of *S. cerevisiae* Puf3 is limited to mitochondrial bio-genesis,~88% of mRNAs with a high-quality motif are genuine targets. The workflow we have used should therefore be informative in defining the regulatory network of other RNA-binding proteins (*Figure 7C*).

How specific mRNAs can be rapidly and specifically targeted for degradation in response to cellu-lar signals remains a major question in understanding eukaryotic gene expression. Our priority model for RNA binding to an adapter protein that acts as a tether to the deadenylation machinery provides new mechanistic insight and brings us closer to being able to predict RNA stability from sequence.

# Materials and methods

## Key resources table

| Reagent type (species) or resource | Designation | Source or reference | Identifiers | Additional information |
|---|---|---|---|---|
| Strain, strain background (*Escherichia coli*) | *E. coli* BL21 star (DE3) | Thermo Fisher Scientific | C601003 | |
| Strain, strain background (*Escherichia coli*) | *E. coli* DH5α (DE3) | Thermo Fisher Scientific | 18258012 | |
| Cell line (*Spodoptera frugiperda*) | *Sf9* | Thermo Fisher Scientific | 11496015 | |
| Peptide, recombinant protein | Ccr4-Not | PMID: 27851962 | | Purified recombinant protein expressed in Sf9 cells following infection with viruse produced from LP_Ccr4Not |
| Peptide, recombinant protein | Ccr4-Not (Ccr4-inactive) | PMID: 27851962 | | Purified recombinant protein with Ccr4(E387) expressed in Sf9 cells following infection with virus produced from LP_P22-60 |
| Peptide, recombinant protein | Ccr4-Not (Caf1-inactive) | PMID: 27851962 | | Purified recombinant protein with Caf1(D53A) expressed in Sf9 cells followinginfection with virus produced from LP_P22-61 |
| Peptide, recombinant protein | MBP-Puf3 | this paper | | Purified recombinant protein expressed in*E. coli* BL21 star (DE3) following transformation with LP_P22-27 |
| Peptide, recombinant protein | MBP-Puf3(Δ63) | this paper | | Purified recombinant protein expressed in*E. coli* BL21 star (DE3) following transformation with LP_P22-28 |
| Peptide, recombinant protein | MBP-Puf3(Δ125) | this paper | | Purified recombinant protein expressed in*E. coli* BL21 star (DE3) following transformation with LP_P22-29 |
| Peptide, recombinant protein | MBP-Puf3(Δ181) | this paper | | Purified recombinant protein expressed in *E. coli* BL2 1 star (DE3) following transformation with LP_P22-30 |

*Continued on next page*

*Continued*

| Reagent type (species) or resource | Designation | Source or reference | Identifiers | Additional information |
|---|---|---|---|---|
| Peptide, recombinant protein | MBP-Puf3(Δ240) | this paper | | Purified recombinant protein expressed in *E. coli* BL21 star (DE3) following transformation with LP_P22-31 |
| Peptide, recombinant protein | MBP-Puf3(Δ300) | this paper | | Purified recombinant protein expressed in *E. coli* BL21 star (DE3) following transformation with LP_P22-32 |
| Peptide, recombinant protein | MBP-Puf3(Δ378) | this paper | | Purified recombinant protein expressed in*E. coli* BL21 star (DE3) following transformation with LP_P22-33 |
| Peptide, recombinant protein | MBP-Puf3(Mut) | this paper | | Purified recombinant protein expressed in*E. coli* BL21 star (DE3) following transformation with LP_P22-34 |
| Peptide, recombinant protein | Puf3(PUM) | this paper | | Purified recombinant protein expressed in *E. coli* BL21 star (DE3) following transformation with LP_P22-35 |
| Peptide, recombinant protein | MBP-Zfs1(FL) | this paper | | Purified recombinant protein expressed in *E. coli* BL21 star (DE3) following transformation with LP_P22-36 |
| Peptide, recombinant protein | MBP-Zfs1(Δ34) | this paper | | Purified recombinant protein expressed in *E. coli* BL21 star (DE3) following transformation with LP_P22-37 |
| Peptide, recombinant protein | MBP-Zfs1(Δ71) | this paper | | Purified recombinant protein expressed in *E. coli*BL21 star (DE3) following transformation with LP_P22-38 |
| Peptide, recombinant protein | MBP-Zfs1(Δ182) | this paper | | Purified recombinant protein expressed in*E. coli* BL21 star (DE3) following transformation with LP_P22-39 |
| Peptide, recombinant protein | MBP-Zfs1(Δ248) | this paper | | Purified recombinant protein expressed in *E. coli* BL21 star (DE3) following transformation with LP_P22-40 |
| Peptide, recombinant protein | MBP-Zfs1(Δ322) | this paper | | Purified recombinant protein expressed in *E. coli* BL21 star (DE3) following transformation with LP_P22-41 |
| Peptide, recombinant protein | MBP-Zfs1(Mut) | this paper | | Purified recombinant protein expressed in *E. coli* BL21 star (DE3) following transformation with LP_P22-42 |
| Peptide, recombinant protein | Zfs1(TZF) | this paper | | Purified recombinant protein expressed in *E. coli* BL21 star (DE3) following transformation with LP_P22-43 |
| Peptide, recombinant protein | Pab1 | PMID: 29932902 | | |
| Commercial assay or kit | Quikchange Lightning Multi Site-Directed Mutagenesis Kit | Agilent Technologies | 210513 | |

*Continued on next page*

*Continued*

| Reagent type (species) or resource | Designation | Source or reference | Identifiers | Additional information |
|---|---|---|---|---|
| Commercial assay or kit | In-Fusion HD Cloning Kit | Takara Bio | 121416 | |
| Commercial assay or kit | Phusion High-Fidelity DNA Polymerase | New England BioLabs | M0530S | |
| Recombinant DNA reagent | MultiBac-Ccr4Not | PMID: 27851962 | LP_Ccr4Not | MultiBac expression vector for *S. pombe* Ccr4-Not |
| Recombinant DNA reagent | MultiBac-Ccr4Not (Ccr4 E387A) | PMID: 27851962 | LP_P22-60 | MultiBac expression vector for *S. pombe* Ccr4-Not (Ccr4 E387A) |
| Recombinant DNA reagent | MultiBac-Ccr4Not(Caf1 D53A) | PMID: 27851962 | LP_P22-61 | MultiBac expression vector for *S. pombe* Ccr4-Not (Caf1 D53A) |
| Recombinant DNA reagent | pMAL-c2x-Puf3(FL) | this paper | LP_P22-27 | pMAL-c2x expression vector for *S. pombe* Puf3(FL) (res1-734) |
| Recombinant DNA reagent | pMAL-c2x-Puf3(Δ63) | this paper | LP_P22-28 | pMAL-c2x expression vector for *S. pombe* Puf3(Δ63) (res63-734) |
| Recombinant DNA reagent | pMAL-c2x-Puf3(Δ125) | this paper | LP_P22-29 | pMAL-c2x expression vector for *S. pombe* Puf3(Δ125) (res125-734) |
| Recombinant DNA reagent | pMAL-c2x-Puf3(Δ181) | this paper | LP_P22-30 | pMAL-c2x expression vector for *S. pombe* Puf3(Δ181) (res181-734) |
| Recombinant DNA reagent | pMAL-c2x-Puf3 (Δ240) | this paper | LP_P22-31 | pMAL-c2x expression vector for *S. pombe* Puf3(Δ240) (res240-734) |
| Recombinant DNA reagent | pMAL-c2x-Puf3 (Δ300) | this paper | LP_P22-32 | pMAL-c2x expression vector for *S. pombe* Puf3(Δ300) (res300-734) |
| Recombinant DNA reagent | pMAL-c2x-Puf3(Δ378) | this paper | LP_P22-33 | pMAL-c2x expression vector for *S. pombe* Puf3(Δ378) (res378-734) |
| Recombinant DNA reagent | pMAL-c2x-Puf3(Mut) | this paper | LP_P22-34 | pMAL-c2x expression vector for *S. pombe* Puf3(Mut) (res1-734; Y449A, Y671A) |
| Recombinant DNA reagent | pGEX6P-2-Puf3(PUM) | this paper | LP_P22-35 | pGEX6P-2 expression vector for *S. pombe* Puf3(PUM) (res378-714) |
| Recombinant DNA reagent | pMAL-c2x-Zfs1(FL) | this paper | LP_P22-36 | pMAL-c2x expression vector for *S. pombe* Zfs1(FL) (res1-404) |
| Recombinant DNA reagent | pMAL-c2x-Zfs1(Δ34) | this paper | LP_P22-37 | pMAL-c2x expression vector for *S. pombe* Zfs1(Δ34) (res34-404) |
| Recombinant DNA reagent | pMAL-c2x-Zfs1(Δ71) | this paper | LP_P22-38 | pMAL-c2x expression vector for *S. pombe* Zfs1(Δ71) (res71-404) |
| Recombinant DNA reagent | pMAL-c2x-Zfs1(Δ182) | this paper | LP_P22-39 | pMAL-c2x expression vector for *S. pombe* Zfs1(Δ182) (res182-404) |
| Recombinant DNA reagent | pMAL-c2x-Zfs1(Δ248) | this paper | LP_P22-40 | pMAL-c2x expression vector for *S. pombe* Zfs1(Δ248) (res248-404) |
| Recombinant DNA reagent | pMAL-c2x-Zfs1(Δ322) | this paper | LP_P22-41 | pMAL-c2x expression vector for *S. pombe* Zfs1(Δ322) (res322-404) |
| Recombinant DNA reagent | pMAL-c2x-Zfs1(Mut) | this paper | LP_P22-42 | pMAL-c2x expression vector for *S. pombe* Zfs1(Mut) (res1-404; F349A, F387A) |
| Recombinant DNA reagent | pGEX6P-2-Zfs1(TZF) | this paper | LP_P22-43 | pGEX6P-2 expression vector for *S. pombe* Zfs1(TZF) (res699-769) |
| Recombinant DNA reagent | pGEX6P-2-Pab1 | PMID: 29932902 | LP_P22-11 | pGEX6P-2 expression vector for *S. pombe* Pab1 (res80-653) |
| Software, algorithm | ImageJ | NIH | https://imagej.nih.gov/ij/ | |

*Continued on next page*

*Continued*

| Reagent type (species) or resource | Designation | Source or reference | Identifiers | Additional information |
|---|---|---|---|---|
| Software, algorithm | GraphPad Prism 6 | GraphPad | https://www.graph pad.com/scientific-software/prism/ | |
| Software, algorithm | switch ANALYSIS | Dynamic Biosensors | https://www.dynamic-biosensors. com/software/ | |
| Software, algorithm | ImageQuant | GE Healthcare | TL 5.2 | |
| Other | Insect-XPRESS protein-free insect cell medium with L-glutamine | Lonza | 12-730Q | |
| Other | Protease Inhibitor Cocktail | Sigma-Aldrich | 11836170001 | |
| Other | Desthiobiotin | IBA | 2-1000-001 | |
| Other | Imidazole | Sigma-Aldrich | I5513 | |
| Other | Formamide | Sigma-Aldrich | 11814320001 | |
| Other | TEMED | Sigma-Aldrich | T9281 | |
| Other | Ammonium persulfate (APS) | Sigma-Aldrich | A3678 | |
| Other | SYBR Safe DNA Gel Stain | Thermo Fisher Scientific | S33102 | |
| Other | SYBR Green II RNA Gel Stain | Thermo Fisher Scientific | S7586 | |
| Other | Ni-NTA Agarose | QIAGEN | 30210 | |
| Other | Glutathione Sepharose 4B | GE Healthcare | 17075601 | |
| Other | Amicon Ultra Centrifugal Filter Units | Millipore | UFC901096 | |
| Other | TWEEN 20 | Sigma-Aldrich | P9416 | |

## Protein purification

Intact Ccr4-Not complex was purified as described previously after overexpression of the seven core subunits of the *Schizosaccharomyces pombe* complex (Ccr4, Caf1, Not1, Not2, Not3, Not4/Mot2 and Rcd1/Caf40) in *Sf*9 cells (*Stowell et al., 2016*). The Caf1-Ccr4 heterodimeric complex was prepared as described previously from the *Sf*9 lysate used for Ccr4-Not expression (*Webster et al., 2018*). *S. pombe* Pab1 (amino acids 80–653) was prepared as described previously (*Webster et al., 2018*).

DNA sequences encoding full-length *S. pombe* Puf3 and Zfs1 were synthesized with codon optimization for *E. coli* (GenScript). Primers Puf3_Fwd and RBP_Rev were used to amplify the Puf3 sequence, and primers Zfs1_Fwd and RBP_Rev were used to amplify the Zfs1 sequence (*Supplementary file 3*). DNA was inserted into pMAL-c2X expression vector using an In-Fusion HD Cloning Kit (Clontech). DNA encoding variants with N-terminal truncations were amplified using primers Puf3_res{63, 125, 181, 240, 300, 378}_Fwd and RBP_Rev for Puf3 and Zfs1_res{34, 71, 182, 248, 322}_Fwd and RBP_Rev for Zfs1. These were also inserted into pMAL-c2X. PCR-based site-directed mutagenesis was performed with a Quikchange Lightning Mutagenesis Kit (Agilent) to generate mutations in the Puf3 RNA-binding domain (Puf3[Mut]: Y449A/Y671A) and the Zfs1 RNA-binding domain (Zfs1[Mut]: F349A/F387A). Primers used to introduce mutations (Puf3_Quikchange and

Zfs1_Quikchange) are listed in **Supplementary file 3**. Sequences of all expression vectors were confirmed by Sanger sequencing with primers pMAL_Seq_Fwd and pMAL_Seq_Rev.

Puf3 and Zfs1 were expressed as N-terminal MBP fusions in BL21 Star (DE3) *E. coli* (Thermo Fisher Scientific). Transformed cells were grown at 37°C to an $A_{600 nm}$ of 0.6 before the temperature was reduced to 18°C and protein expression induced by the addition of IPTG to 0.5 mM. Growth was continued for 18 hr before cells were harvested by centrifugation and flash frozen for storage at −80°C. Cells were defrosted and lysed by sonication in lysis buffer (50 mM Tris-HCl pH 8, 250 mM KCl, 1 mM TCEP) supplemented with protease inhibitor cocktail (Roche) and DNase I (5 µg/ml) (Sigma). The lysate was cleared by centrifugation at 40,000 × g for 30 min at 4°C. Supernatant was applied to amylose affinity resin (1 ml bed volume per 2 l culture; New England Biolabs), and incubated with rotation for 1 hr at 4°C. Resin was separated from lysate with an Econo-Column (Bio-Rad) and washed three times with 50 ml of RBP buffer (20 mM Tris-HCl pH 8, 50 mM NaCl, 0.1 mM TCEP). Protein was eluted from the resin with RBP buffer supplemented with 50 mM maltose. Sample was loaded onto a HiTrap Q HP 5 ml column (GE Healthcare), and the protein was eluted with a gradient of 20 column volumes into RBP buffer containing 1 M NaCl. Protein in the elution peak was concentrated to 0.5 ml with an Amicon Ultra concentrator (4 ml, 50 kDa MWCO, Millipore) by centrifugation at 3000 × g at 4°C. Sample was then applied to a Superdex 200 10/300 GL size-exclusion column (GE Healthcare) equilibrated in RBP buffer. The column was run at 0.5 ml/min at 4°C. Full-length protein eluted at 10–13 ml, typically together with smaller protein species identified by tandem mass spectrometry to be truncated protein lacking C-terminal residues. The sample was applied to a HiTrap Heparin HP 5 ml column (GE Healthcare), and the protein was eluted with a gradient of 20 column volumes into RBP buffer containing 1 M NaCl. Full-length protein eluted at ~250 mM NaCl, while shorter fragments eluted at 100–200 mM NaCl. Peak fractions were concentrated with an Amicon Ultra 50 kDa MWCO centrifugal concentrator (Millipore) and stored at −80°C.

The RNA-binding PUM domain of *S. pombe* Puf3 was defined as residues 378–714, while the RNA-binding TZF domain of *S. pombe* Zfs1 was defined as residues 699–769. DNA sequences encoding the RNA-binding domains of Puf3 (Puf3$^{PUM}$) and Zfs1 (Zfs1$^{TZF}$) were amplified from vectors encoding full-length proteins using primers Puf3_PUM_Fwd and Puf3_PUM_Rev or Zfs1_TZF_Fwd and Zfs1_TZF_Rev (**Supplementary file 3**). These were inserted into pGEX-6P-2 expression vector using an In-Fusion HD Cloning Kit (Clontech). Puf3$^{PUM}$ and Zfs1$^{TZF}$ were expressed as N-terminal GST fusions in BL21 Star (DE3) *E. coli* (Thermo Fisher Scientific). Expression and cell lysis were performed as described for full-length MBP-Puf3 and MBP-Zfs1. Clarified lysate was applied to glutathione sepharose 4B affinity resin (1 ml bed volume per 2 l culture; New England Biolabs), and incubated with rotation for 1 hr at 4°C. Resin was separated from lysate with an Econo-Column (Bio-Rad) and washed three times with 50 ml of RBP buffer. Protein was eluted from the resin in 20 ml of glutathione elution buffer (20 mM HEPES pH 7.4, 150 mM NaCl, 0.1 mM TCEP, 50 mM glutathione). The GST tag was cleaved by incubation with 1 mg of 3C protease for 16 hr at 4°C. Cleaved protein was buffer-exchanged into 20 mM Bis-Tris pH 6.5, 50 mM NaCl, 0.1 mM TCEP with a HiPrep 26/10 desalting column (GE Healthcare). The sample was applied to a Mono S 5/50 GL cation-exchange column (GE Healthcare) and eluted over 20 column volumes into buffer supplemented to 1 M NaCl. Pure protein eluted at ~200 mM NaCl, and samples were pooled and stored at −80°C.

## Deadenylation assays

Deadenylation activity was measured as previously described (*Webster et al., 2017*). Puf3 and Zfs1 were prepared in 20 mM HEPES pH 7.4, 50 mM NaCl, 0.1 mM TCEP and added to a final concentration of 250 nM in the reaction, unless otherwise indicated. RBPs were incubated with RNA for 10 min at 22°C prior to the addition of enzyme to allow protein-RNA binding to reach equilibrium. Puf3-target-A30 and Zfs1-target-A30 RNA were synthesized with a 5' 6-FAM fluorophore label (Integrated DNA Technologies). The RNAs were designed with a consensus Puf3 or Zfs1 binding site ending six nucleotides upstream of a poly(A) tail, within a 23 nt non-poly(A) sequence. 200 nM RNA was used in each reaction. Ccr4-Not and Caf1-Ccr4 were prepared at 1 µM (10×) in 20 mM HEPES pH 7.4, 400 mM NaCl, 2 mM Mg(OAc)$_2$, 0.1 mM TCEP and added to a final concentration of 100 nM in the reaction. Reactions were performed in 20 mM PIPES pH 6.8, 10 mM KCl, 45 mM NaCl, 2 mM Mg(OAc)$_2$, 0.1 mM TCEP (includes components added with protein factors) at 22°C.

Reactions were stopped at the indicated time points by addition of 2 × denaturing loading dye (95% formamide, 10 mM EDTA, 0.01% w/v bromophenol blue). Samples were applied to TBE (Tris-borate-EDTA), 20% polyacrylamide gels containing 7 M urea and run at 400 V in 1 × TBE running buffer. Gels were scanned with a Typhoon FLA-7000 with excitation at 478 nm. Densitometric analysis was performed with *ImageJ* as described previously (*Schneider et al., 2012*; *Webster et al., 2017*). Poly(A) tail lengths were calibrated using RNA markers with no tail and tails of known length. Intermediate tail lengths were calculated by counting bands on gels with single-nucleotide resolution as described previously (*Webster et al., 2017*). Deadenylation rates were calculated from a plot of the size of the most abundant RNA species versus time. Statistical significance was calculated by one-way ANOVA in GraphPad Prism.

For RNA competition experiments, Zfs1-target-A30 and *Sc*Puf3(5U)-target-A30 RNAs were synthesized with a 5'-Alexa647 fluorophore label, and Puf5-target-A30 with a 5'-Cy3 fluorophore label (Integrated DNA Technologies). Reactions contained 100 nM of each RNA, mixed prior to addition of protein factors. Gels were scanned twice with a Typhoon FLA-7000 with excitation at 478 nm (fluorescein detection) and either 633 nm (Alexa647 detection) or 573 nm (Cy3 detection). Images were superimposed with color overlay.

## Electrophoretic mobility shift assays

Binding reactions (10 µl) were prepared by adding protein at the indicated concentration to RNA (200 nM) in 20 mM PIPES pH 6.8, 10 mM KCl, 90 mM NaCl, 2 mM Mg(OAc)$_2$, 0.1 mM TCEP. For RNA competition experiments, 100 nM of each RNA was mixed prior to addition of protein. The sample was incubated for 15 min at 22°C before the addition of 6 × loading dye (30% glycerol and 0.2% w/v orange G). Samples were loaded onto 6% TBE-polyacrylamide non-denaturing gels and electrophoresis was performed at 100 V in 1 × TBE running buffer. Gels were scanned with a Typhoon FLA-7000 with excitation at 478 nm (fluorescein detection) and 633 nm (Alexa647 detection).

## Fluorescence polarization assays

A two-fold protein dilution series was prepared in 20 mM HEPES pH 7.4, 150 mM NaCl. Proteins were incubated for 2 hr at 22°C with 0.2 nM 5' 6-FAM RNA (synthesized by IDT). Fluorescence polarization was measured with a PHERAstar Plus microplate reader (BMG Labtech). Dissociation constants were estimated by non-linear regression in *GraphPad Prism 6*. Error bars indicate the standard deviation in three replicate measurements.

## SwitchSENSE kinetic analysis

Kinetic measurements were performed using a DRX series instrument with a MPC-48–2-Y1 chip (Dynamic Biosensors). Hybrid oligonucleotides were synthesized (IDT) with the RNA of interest (*Table 1*) at the 5' end followed by single-stranded DNA complementary in sequence to the fluorescently labelled oligonucleotide on the chip. Annealing was performed by flowing 500 nM oligonucleotide over the chip for 4 min in a buffer of 20 mM HEPES pH 7.4, 40 mM NaCl and 0.001% Tween-20.

Purified Puf3[PUM] (residues 378–714) was applied to the chip in binding buffer (20 mM HEPES pH 7.4, 150 mM NaCl and 0.001% Tween-20) at 20°C with a flow rate of 30 µl/min. For association measurements, a series of protein concentrations was tested: 5, 10, 15, 20, 25, 30, 50, 70 nM for *Sc*Puf3- and *Sc*Puf5-target RNA; 50, 100, 150, 200 nM for poly(A). The dynamic response represents the change in nanolever switch speed on the timescale 0–4 µsec as described previously (*Langer et al., 2013*). The observed association rate ($K_{obs}$) was estimated by fit of an exponential curve to data points collected with a sampling rate of 1 s with *GraphPad Prism 6*. Error bars indicate the standard error in three replicate measurements. Association rate ($k_{on}$) was calculated by linear regression analysis of the protein concentration series.

For analysis of dissociation rate, 20 nM Puf3[PUM] was loaded onto RNA as described above. Binding buffer was then flowed across the chip at 20°C with a flow rate of 30 µl/min. Data points from dissociation experiments with *Sc*Puf3-target and *Sc*Puf5-target RNAs were averaged in 10 s intervals to improve the signal-to-noise. Dissociation rates ($k_{off}$) were estimated by fitting of an exponential function with *GraphPad Prism 6*.

**Table 1.** Sequences of RNAs used in this study.

| Name | Sequence | Experiment |
|---|---|---|
| Zfs1 target A30 | AAUCAUCCUUAUUUAUUACCAUU[A]30 | Deadenylation |
| Puf3 target A30 / Puf3 target (5A) A30 | AACUGUUCCUGUAAAUACGCCAG[A]30 | Deadenylation |
| Puf3 target (5U) A30 | AACUGUUCCUGUAUAUACGCCAG[A]30 | Deadenylation |
| Puf5 target | AACUGUUGUUGUAUGUACGCCAG[A]30 | Deadenylation |
| ARE/Zfs1 target | AUUAUUUAUU | Fluorescence polarization |
| PRE/Puf3 target (5A) | ACCUGUAAAUA | Fluorescence polarization |
| Puf3 target (5U) | ACCUGUAUAUA | Fluorescence polarization |
| Puf5 target | AGUUGUAUGUA | Fluorescence polarization |
| A30 | [A]30 | Fluorescence polarization |
| Puf3 target (5A) | ACCUGUAAAUAGGCG-[*DNA] | SwitchSENSE |
| Puf3 target (5U) | ACCUGUAUAUAGGCG-[*DNA] | SwitchSENSE |
| Puf5 target | AGUUGUAUGUAGGCG-[*DNA] | SwitchSENSE |
| Poly(A) | [A]15-[*DNA] | SwitchSENSE |

*DNA for hybridization to nanolever:
ATCAGCGTTCGATGCTTCCGACTAATCAGCCATATCAGCTTACGACTA
DOI: https://doi.org/10.7554/eLife.40670.015

## Bioinformatic analysis of ScPuf3 mRNA targets

*Saccharomyces cerevisiae* mRNAs that are candidates for Puf3 regulation were collated from published sources (*Gerber et al., 2004*; *Kershaw et al., 2015*; *Lapointe et al., 2015*). Proportional Venn diagram of gene overlap was generated with Biovenn (*Hulsen et al., 2008*) and redrawn for publication. 3'-UTR sequences were defined using poly(A) sites previously identified (*Ozsolak et al., 2010*). The most statistically enriched RNA motif from the combined set of 1331 mRNAs was identified with MEME (*Bailey et al., 2015*). Analysis of sequence context around the target motifs (10 nucleotides in each direction) was performed with WebLogo 3 (*Crooks et al., 2004*).

Motifs were identified in mRNA sequences with *Geneious* v 7.0.6 (*Kearse et al., 2012*) with the following general expressions: motif 1 (CNUGUAAAUA), motif 2 (CNUGUANAUA), motif 3 (UGUAAAUA) and motif 4 (UGUAAAUA). Sequences within motif category 1 are by definition within 2, 3 and 4. Once it was evident that mRNAs with motif 1 were significantly enriched in *Sc*Puf3 relative to the remainder of mRNAs in categories 2, 3 and 4, they were assigned to a distinct category. No significant differences in *Sc*Puf3 enrichment were detected between mRNAs with other permutations not presented (−2: A, U or G; −1 any nucleotide;+5 C, G or U). For this reason, these sequences were treated as equivalent and the mRNAs containing them analyzed as a group. We do not exclude the possibility that differences in *Sc*Puf3 binding exist because statistically evaluation is limited to the number of endogenous genes containing each permutation. Motifs were assigned to the category 3'-UTR if at least part of the motif contained nucleotides 3' of the coding sequence (including the stop codon). Statistical significance was evaluated with a Mann-Whitney rank comparison test in GraphPad Prism.

Gene ontology (GO) analysis was performed with PANTHER over-representation test using the dataset released on 2018-02-02 (*Mi et al., 2017*). Comprehensive GO term analysis of significant over-representation is available in *Supplementary file 2*. Detailed analysis of individual gene function was performed with the Saccharomyces Genome Database (http://yeastmine.yeastgenome.org) and the references contained therein.

## Acknowledgements

We thank Terence Tang for comments on the manuscript; Jianguo Shi for assistance with baculovirus; and Stephen McLaughlin and Chris Johnson for assistance and advice with biophysical techniques. This work was supported by the European Research Council under the European Union's

Seventh Framework Programme (FP7/2007–2013)/ERC Starting grant no. 261151 (to LAP); under the European Union's Horizon 2020 research and innovation programme (ERC Consolidator grant agreement No 725685) (to LAP); and Medical Research Council (MRC) grant MC_U105192715 (LAP).

## Additional information

### Funding

| Funder | Grant reference number | Author |
| --- | --- | --- |
| Medical Research Council | MC_U105192715 | Lori A Passmore |
| Horizon 2020 Framework Programme | 725685 | Lori A Passmore |
| Seventh Framework Programme | 261151 | Lori A Passmore |

The funders had no role in study design, data collection and interpretation, or the decision to submit the work for publication.

### Author contributions

Michael W Webster, Conceptualization, Data curation, Formal analysis, Investigation, Methodology, Writing—original draft; James AW Stowell, Resources, Formal analysis, Investigation, Writing—review and editing; Lori A Passmore, Conceptualization, Supervision, Funding acquisition, Writing—original draft

### Author ORCIDs

Lori A Passmore (iD) https://orcid.org/0000-0003-1815-3710

### Decision letter and Author response

Decision letter https://doi.org/10.7554/eLife.40670.021
Author response https://doi.org/10.7554/eLife.40670.022

## Additional files

### Supplementary files

• Supplementary file 1. Analysis of Puf3-motif quality in putative mRNA targets of Puf3. This table includes the *S. cerevisiae* mRNAs that are candidates for Puf3 regulation, obtained from published sources as detailed in Materials and methods. The highest quality motif present in each mRNA is indicated with the category numbers defined in *Figure 6B* (1–4), and 5 for Δcore. Functional categories were defined in this study based on published data obtained from the SGD database.
DOI: https://doi.org/10.7554/eLife.40670.016

• Supplementary file 2. Gene Ontology analysis of mRNA groups defined by Puf3-motif quality and location. Statistically significant gene ontology over-representation was identified with PANTHER as detailed in Materials and methods. Terms were identified for the mRNA groups defined by the presence of motifs 1, 2 or 3 within the 3' UTR (see Figutre 6B for motif quality definitions). No significantly enriched GO terms were identified for motif 4 in the 3' UTR or for any groups with motifs in the coding sequence. Significantly enriched GO terms for the mRNAs defined by a mismatch within the seven core nucleotides (Δcore) were identified and are shown here.
DOI: https://doi.org/10.7554/eLife.40670.017

• Supplementary file 3. Plasmids and Oligonucleotides. Plasmids and oligos used in this study are listed.
DOI: https://doi.org/10.7554/eLife.40670.018

• Transparent reporting form
DOI: https://doi.org/10.7554/eLife.40670.019

## Data availability

All data generated during this study are included in the manuscript and supporting files. Source data are provided in Supplementary Files 1 and 2.

The following datasets were generated:

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
