## [Decision Letter]

Thank you for submitting your article "The mechanism of Ccr4-Not recruitment to specific mRNAs involves sequence-selective tethering by RNA-binding proteins" for consideration by *eLife*. Your article has been reviewed by three peer reviewers, including Rachel Green as the Reviewing Editor and Reviewer #1, and the evaluation has been overseen by Kevin Struhl as the Senior Editor.

We have received reviewer comments and have discussed the manuscript together in coming to a decision. Overall, all three reviewers felt the manuscript detailing how RNA binding protein promote Ccr4-Not mediated deadenylation using a really beautiful in vitro reconstituted system was extremely well executed and brings to light important features of this biological interaction. Importantly, the dependence of the rates of deadenylation on the dissociation rates of the complexes is interesting, and nicely complements previously published in vivo work on this topic (Lapointe et al., 2017). The analysis of already published *S. cerevisiae* CLIP data is seen to be strength, but the discussion needs some re-working to make connections between the in vitro and in vivo work clearer, and to acknowledge previous in vivo studies in the field that came to related conclusions.

We recommend that the manuscript be revised, taking into account the reviewers’ suggestions, and that it will then be appropriate for publication in *eLife*.

*Reviewer #1:*

In this manuscript by Webster et al., the authors describe experiments that define how Ccr4-Not-promoted deadenylation is promoted by RNA binding proteins that recognize specific motifs on the targeted mRNAs. The experiments are entirely performed in vitro with purified proteins and clean kinetically-refined deadenylation assays. The studies show that the fission yeast protein Zfs1 and Puf3 increase the rates of deadenylation on mRNAs that contain a motif that they are known to recognize, and that this recruitment is primarily mediated by the very long disordered regions of these RNA binding proteins (in some opposition to earlier work from Goldstrohm's group). The specificity of the RNA binding protein is shown to be critical, but when concentrations become high, less optimal mRNA targets readily become targets (since binding affinities for optimal and non-optimal sequences only differ by about 100-fold). In the final section of the manuscript, the authors perform a detailed analysis of three different data sets in the literature defining the targets of these mRNA binding proteins in genome-wide experiments. Here the authors are able to better define that specificities of these proteins, and for Puf3, to rationalize how this protein is principally involved in regulating the stability of mRNAs associated with mitochondrial function. This final analysis lends some in vivo depth to the manuscript that raised its significance substantially. In general, the manuscript was very clearly written and the experiments nicely supported the stated conclusions. I recommend the paper for publication in *eLife*.

*Reviewer #2:*

This manuscript presents biochemical evidence that the *S. pombe* Pumilio and Tristetraprolin homologs recruit the Ccr4-Not complex to deadenylate mRNAs.

While the biochemical data presented in this manuscript are of high quality, evidences that Pumilio and Tristetraprolin recruit the Ccr4-Not complex to deadenylate mRNAs in other species have been widely documented. In this context, the data described in this manuscript appear mostly confirmatory, even if they are based on a different, more direct, analysis. The conceptual advance brought by this manuscript is thus limited.

While the manuscript does bring new details, e.g. on the context effect on RNA binding, those are not unexpected and will be of interest only to specialists. Moreover, in some case, specific conclusions remain preliminary (e.g. the region of *S. pombe* Tristetraprolin interacting with the Ccr4-Not complex is not as well defined as the one described 5 years ago for its human homolog for which structural data were presented). Furthermore, a significant fraction of the manuscript focuses on analysis of published data emanating from *S. cerevisiae* that are rather descriptive, not validated, and poorly connected to the experimental data presented in the rest of the manuscript.

Altogether, *eLife* cannot publish all manuscripts that it receives for review and, as a broad journal, should focus on articles of wide interest. In this specific case, given the background knowledge available in the current literature, it is difficult to argue why it should be published in *eLife*. Indeed, these data will be of interest mostly to a group of specialists. I recommend publication in a journal targeting a more restricted audience.

*Reviewer #3:*

Webster et al. is an important contribution to understanding mechanisms of mRNA control. The use of purified, reconstituted Ccr4-Not and two different RNA-binding proteins, and the care taken in experimental design, as in the dual reporter assays (Figures 3 and 4), make the data and conclusions compelling.

1) A central conclusion is that dissociation rates are a key contributor to rates of deadenylation. This important conclusion complements in vivo "tagging" experiments on Puf3^PUM^ (Lapointe et al., 2017). In those studies, the number of uridines added in tagging very likely is related to PUF-RNA dissociation rates in vivo, which correlates well with whether the mRNAs are in fact regulated by Puf3 in the cell. It is in the authors’ interest to make the point that in vivo studies of tagging led to the suggestion of "sampling" vs. "biological regulation" as a function of off-rates; the convergence of the in vitro and in vivo data is striking.

2) The experiments use *S. pombe* Puf3^PUM^ and *S. pombe* Ccr4-Not. This raises a potential confusion for the reader, in that the computational analyses done later in the paper focus on mRNAs that are bound and controlled in vivo in *S. cerevisiae* by the *S. cerevisiae* protein. Unfortunately, this makes some of the discussion of Figure 6 a bit confusing, and needs clarification in the text. It is not that mRNAs that bind *S. pombe* Puf3 in *S. pombe* necessarily are mitochondrial in function, a point that is relevant to computational studies of site divergence during evolution (Jiang et al., 2012, Mol. Biol. Evol.; Hogan et al., 2015; Wilinski et al., 2017).

The data and analyses of Figure 6 certainly are informative, and the issue should be dealt with by clarifications in the Results and Discussion sections.

3) I may have missed this, but how do the authors quantify deadenylation "rate" (Figure 1—figure supplement 1, Figure 2B)? Does "rate" refer to moles of adenosine monophosphate released per unit time, or the amount of fully deadenylated product per unit time, or something else?

[Editors' note: further revisions were requested prior to acceptance, as described below.]

Thank you for resubmitting your work entitled "The mechanism of Ccr4-Not recruitment to specific mRNAs involves sequence-selective tethering by RNA-binding proteins" for further consideration at *eLife*. Your revised article has been favorably evaluated by Kevin Struhl (Senior Editor), a Reviewing Editor, and two reviewers.

Overall, the reviewers felt that most comments had been adequately addressed. However, one reviewer feels that the title and Abstract of the manuscript must be altered to acknowledge what precisely is novel in the study which is the biochemical discrimination between target and non-target mRNAs. Once this issue has been addressed the manuscript should be ready for publication in *eLife*.

*Reviewer #2:*

Passmore and colleagues have revised their manuscript, considering most referees' comments. Minor points have been clarified. In line with editorial recommendation, authors argue that their "major conceptual advance is not that Pumilio and Tristetraprolin proteins promote deadenylation, but an improved understanding of two aspects of this process – (1) how RNA-binding proteins distinguish between target and non-target mRNAs, and (2) how RNA-binding proteins recruit Ccr4-Not to achieve rapid deadenylation".

As discussed earlier, our new understanding of interaction of RNA binding proteins with the Ccr4-Not is limited in the context of data already available in the literature (see below). Consistently, the authors indicate that a full understanding of this binding will require further studies. This point remains accessory.

Altogether, there is a general agreement that the main new point addressed in this manuscript is how RNA-binding proteins distinguish between target and non-target mRNAs. While the text has been improved to clarify this issue, the manuscript title "The mechanism of Ccr4-Not recruitment to specific mRNAs involves sequence-selective tethering by RNA-binding proteins" and Abstract are still confusing. They don't do justice to the numerous manuscripts and groups that have reported over the last 15 years that a wide range of RNA binding proteins recruit the Ccr4-Not complex to target mRNAs, in several cases in the recent years including structural details (a non-exhaustive list includes Semotok et al., 2005; Finoux et al., 2006; Goldstrohm et al., 2006; Chicoine et al., 2007; Goldstrohm et al., 2007; Rendl et al., 2008; Suzuki et al., 2010; Braun et al., 2011; Van Etten et al., 2012; Fabian et al., 2013; Bandhari et al., 2014; Raisch et al., 2016; Sgromo et al., 2017; Bulbrook et al., 2018; Sgromo et al., 2018…). The concept of tethering of the Ccr4-Not complex by RNA binding proteins appears without contest well established.

Before publication, I recommend the title to be modified to reflect the new findings reported in the manuscript allowing RNA binding proteins to differentiate target and non-target sequences. The Abstract would also merit clarification.

*Reviewer #3:*

The authors have adequately addressed my comments.

---

## [Author Response]

We recommend that the manuscript be revised, taking into account the reviewers’ suggestion, and that it will then be appropriate for publication in eLife.Reviewer #2:This manuscript presents biochemical evidence that the S. pombe Pumilio and Tristetraprolin homologs recruit the Ccr4-Not complex to deadenylate mRNAs.While the biochemical data presented in this manuscript are of high quality, evidences that Pumilio and Tristetraprolin recruit the Ccr4-Not complex to deadenylate mRNAs in other species have been widely documented. In this context, the data described in this manuscript appear mostly confirmatory, even if they are based on a different, more direct, analysis. The conceptual advance brought by this manuscript is thus limited.

In this work, we examined the mechanisms that allow RNA-binding proteins to achieve sequence-selective deadenylation. The major conceptual advance is not that Pumilio and Tristetraprolin proteins promote deadenylation, but an improved understanding of two aspects of this process – (1) how RNA-binding proteins distinguish between target and non-target mRNAs, and (2) how RNA-binding proteins recruit Ccr4-Not to achieve rapid deadenylation.

The RNA-binding proteins Puf3 and Zfs1 from *S. pombe* were selected for this study precisely because homologs from other organisms have been documented to recruit Ccr4-Not. They are therefore a useful model system to examine the biochemical mechanisms of RNA-binding proteins that recruit Ccr4-Not.

While the manuscript does bring new details, e.g. on the context effect on RNA binding, those are not unexpected and will be of interest only to specialists. Moreover, in some case, specific conclusions remain preliminary (e.g. the region of S. pombe Tristetraprolin interacting with the Ccr4-Not complex is not as well defined as the one described 5 years ago for its human homolog for which structural data were presented).

The evidence we present indicates that the kinetics of binding define which RNA sequence motifs are recognized as targets by RNA-binding proteins. We describe this as a ‘priority model’ of RNA binding, which challenges the traditional distinction between specific and non-specific binding. This model is of general importance. By defining the RNA motifs recognized by *S. cerevisiae* Puf3 in a priority order (categories 1 – 4 in Figure 6), we were able to ascertain an improved definition of its mRNA targets, and consequently its cellular role. While we have used Pumilio proteins as a model system, this method of analysis could be applied to any RNA-binding protein.

We did not attempt to define the interaction interface of RNA-binding proteins and Ccr4-Not at the resolution of amino acids. Instead, we show that multiple regions within the low-complexity regions of *S. pombe* Puf3 and Zfs1 contribute to Ccr4-Not recruitment, where each individual binding site is likely of relatively low affinity. (The interaction of TTP with CNOT1 described by Fabian et al. has an affinity (*K_D_*) of ~2 µM. Other parts of TTP are likely to contribute to the interaction with Ccr4-Not). It is likely that these RNA-binding proteins interact with Ccr4-Not through an avidity driven mechanism, and a full understanding of this distributed binding will require studies of larger parts of the proteins.

Furthermore, a significant fraction of the manuscript focuses on analysis of published data emanating from S. cerevisiae that are rather descriptive, not validated, and poorly connected to the experimental data presented in the rest of the manuscript.

Our analysis of published experimental data served to test whether our model of RNA-binding is important in vivo: do subtle differences in motif quality dictate which mRNAs are targets of Puf3? The ability to predict mRNA targets based only on RNA motif quality had not previously been tested. Our analysis of published experimental data revealed that examining subtle differences in motif quality has substantial predictive power.

Reviewer #3:Webster et al. is an important contribution to understanding mechanisms of mRNA control. The use of purified, reconstituted Ccr4-Not and two different RNA-binding proteins, and the care taken in experimental design, as in the dual reporter assays (Figures 3 and 4), make the data and conclusions compelling.1) A central conclusion is that dissociation rates are a key contributor to rates of deadenylation. This important conclusion complements in vivo "tagging" experiments on Puf3^PUM^ (Lapointe et al., 2017). In those studies, the number of uridines added in tagging very likely is related to PUF-RNA dissociation rates in vivo, which correlates well with whether the mRNAs are in fact regulated by Puf3 in the cell. It is in the author's interest to make the point that in vivo studies of tagging led to the suggestion of "sampling" vs. "biological regulation" as a function of off-rates; the convergence of the in vitro and in vivo data is striking.

In the Discussion we have added the text:

“An indication ofin vivo dissociation rates of *Sc*Puf3 were previously obtained using RNA tagging (Lapointe et al., 2015). […] Taken together, there is a striking convergence between the results of in vitro(this study) and in vivo(Lapointe et al., 2015) studies.”

2) The experiments use S. pombe Puf3^PUM^ and S. pombe Ccr4-Not. This raises a potential confusion for the reader, in that the computational analyses done later in the paper focus on mRNAs that are bound and controlled in vivo in S. cerevisiae by the S. cerevisiae protein. Unfortunately, this makes some of the discussion of Figure 6 a bit confusing, and needs clarification in the text. It is not that mRNAs that bind S. pombe Puf3 in S. pombe necessarily are mitochondrial in function, a point that is relevant to computational studies of site divergence during evolution (Jiang et al., 2012, Mol. Biol. Evol.; Hogan et al., 2015; Wilinski et al., 2017).The data and analyses of Figure 6 certainly are informative, and the issue should be dealt with by clarifications in the Results and Discussion sections.

We have changed our nomenclature of budding yeast Puf3 to *Sc*Puf3. This clarifies the text, making it easier to discriminate whether we are discussing budding yeast or fission yeast proteins.

We also now state in the Introduction:

“Puf3 homologs in other organisms (including *S. pombe*) do not necessarily regulate mitochondrial function (Hogan et al., 2015).”

In addition, in the Results, we have changed the text to include the following:

“The ability of Puf3 to discriminate between similar sequences is likely to play a role in defining the mRNAs within its regulatory network. […] Thus, we applied our in vitro findings using *S. pombe* proteins to a computational analysis of *S. cerevisiae* Puf3.”

3) I may have missed this, but how do the authors quantify deadenylation "rate" (Figure 1—figure supplement 1, Figure 2B)? Does "rate" refer to moles of adenosine monophosphate released per unit time, or the amount of fully deadenylated product per unit time, or something else?

All rates were calculated using densitometric analysis of the signal from 5′-fluorophore labeled RNAs in denaturing polyacrylamide gels that have single-nucleotide resolution. We determined the poly(A) tail length of the most abundant RNA species at each time point and plotted this versus time. The slope is then the average deadenylation rate. The rate is therefore the number of adenosines removed per unit time. This method is described in detail in the reference given (Webster et al., 2017). We now add more description in the Materials and methods.

In Figure 2B we state in the legend:

“The most abundant RNA size decreased linearly with time, and average reaction rates in number of adenosines removed/min were determined by linear regression on triplicate measurements.”

This is the same method used in Figure 1—figure supplement 1E and Figure 2—figure supplement 1C, and we have now added this text to those figure legends.

[Editors' note: further revisions were requested prior to acceptance, as described below.]

Reviewer #2:Passmore and colleagues have revised their manuscript, considering most referees' comments. Minor points have been clarified. In line with editorial recommendation, authors argue that their "major conceptual advance is not that Pumilio and Tristetraprolin proteins promote deadenylation, but an improved understanding of two aspects of this process – (1) how RNA-binding proteins distinguish between target and non-target mRNAs, and (2) how RNA-binding proteins recruit Ccr4-Not to achieve rapid deadenylation".As discussed earlier, our new understanding of interaction of RNA binding proteins with the Ccr4-Not is limited in the context of data already available in the literature (see below). Consistently, the authors indicate that a full understanding of this binding will require further studies. This point remains accessory.Altogether, there is a general agreement that the main new point addressed in this manuscript is how RNA-binding proteins distinguish between target and non-target mRNAs. While the text has been improved to clarify this issue, the manuscript title "The mechanism of Ccr4-Not recruitment to specific mRNAs involves sequence-selective tethering by RNA-binding proteins" and Abstract are still confusing. They don't do justice to the numerous manuscripts and groups that have reported over the last 15 years that a wide range of RNA binding proteins recruit the Ccr4-Not complex to target mRNAs, in several cases in the recent years including structural details (a non-exhaustive list includes Semotok et al., 2005; Finoux et al., 2006; Goldstrohm et al., 2006; Chicoine et al., 2007; Goldstrohm et al., 2007; Rendl et al., 2008; Suzuki et al., 2010; Braun et al., 2011; Van Etten et al., 2012; Fabian et al., 2013; Bandhari et al., 2014; Raisch et al., 2016; Sgromo et al., 2017; Bulbrook et al., 2018; Sgromo et al., 2018…). The concept of tethering of the Ccr4-Not complex by RNA binding proteins appears without contest well established.

We certainly did not intend to diminish previous data that show many details of the interactions of RNA-binding proteins with Ccr4-Not. We cited many of the papers mentioned by the reviewer and discuss these data many times throughout the manuscript. We include descriptions of these in the Introduction and Discussion. In addition, in the Results section when we first describe accelerated deadenylation, we state: “Therefore, like metazoan homologs, *S. pombe* Puf3 and Zfs1 bind directly to Ccr4-Not.”

To avoid confusion, we made a change to the Results section to further emphasise previous results:

“As mentioned above, specific interactions between Ccr4-Not and Pumilio/TTP proteins had been previously described (Fabian et al., 2013; Goldstrohm et al., 2006; Van Etten et al., 2012). To dissect the mechanism of accelerated deadenylation and to determine the contributions of different parts of Puf3 and Zfs1, we made a series of mutations and truncations in these proteins, and tested their abilities to stimulate deadenylation.”

Before publication, I recommend the title to be modified to reflect the new findings reported in the manuscript allowing RNA binding proteins to differentiate target and non-target sequences. The Abstract would also merit clarification.

The title we chose (“The mechanism of Ccr4-Not recruitment to specific mRNAs involves sequence-selective tethering by RNA-binding proteins”) was meant to emphasize the sequence-selective nature of the RNA-binding proteins. In other words, the RNA-binding proteins that interact with Ccr4-Not can differentiate target and non-target sequences.

We have now changed the title to: “RNA-binding proteins distinguish between similar sequence motifs to promote targeted deadenylation by Ccr4-Not.”

We also modified the Abstract in the revised manuscript.